# The impact of the genetic background on gene deletion phenotypes in *Saccharomyces cerevisiae*

Marco Galardini[1,†] (ID), Bede P Busby[1,2,†] (ID), Cristina Vieitez[1,2], Alistair S Dunham[1], Athanasios Typas[2,*] (ID) & Pedro Beltrao[1,**] (ID)

## Abstract

Loss-of-function (LoF) mutations associated with disease do not manifest equally in different individuals. The impact of the genetic background on the consequences of LoF mutations remains poorly characterized. Here, we systematically assessed the changes in gene deletion phenotypes for 3,786 gene knockouts in four *Saccharomyces cerevisiae* strains and 38 conditions. We observed 18.5% of deletion phenotypes changing between pairs of strains on average with a small fraction conserved in all four strains. Conditions causing higher wild-type growth differences and the deletion of pleiotropic genes showed above-average changes in phenotypes. In addition, we performed a genome-wide association study (GWAS) for growth under the same conditions for a panel of 925 yeast isolates. Gene–condition associations derived from GWAS were not enriched for genes with deletion phenotypes under the same conditions. However, cases where the results were congruent indicate the most likely mechanism underlying the GWAS signal. Overall, these results show a high degree of genetic background dependencies for LoF phenotypes.

**Keywords** chemical genomics; genetic background; LoF phenotypes; *Saccharomyces cerevisiae*

**Subject Categories** Chromatin, Transcription & Genomics; Genetics, Gene Therapy & Genetic Disease

**Mol Syst Biol. (2019) 15: e8831**

## Introduction

While a mutation can be associated with specific disorders, it has long been observed that not all individuals carrying the disease variant will manifest it. Even for diseases caused by mutations in a single gene (i.e. monogenic disorders), incomplete penetrance is frequent, presumably due to differences in the genetic background (Kammenga, 2017; Hou *et al*, 2018). Modulators of penetrance of disease-causing variants have been identified for many human diseases (Cohen *et al*, 2005; Flannick *et al*, 2014; Chen *et al*, 2016) and loss-of-function (LoF) mutations in different model organisms (Hamilton & Yu, 2012; Chari & Dworkin, 2013; Vu *et al*, 2015; Chow *et al*, 2016; Mullis *et al*, 2018). This impact of the genetic background on the phenotypic consequence of LoF mutations affects our ability to predict phenotypes based on genetic variants. In *Saccharomyces cerevisiae* and *E. coli*, gene deletion phenotypes have been extensively measured for all genes across hundreds of stress conditions (Hillenmeyer *et al*, 2008; Nichols *et al*, 2011). However, genes carrying putative LoF mutations in different strains are only weakly predictive of expected gene deletion phenotypes (Jelier *et al*, 2011; Galardini *et al*, 2017; Wagih *et al*, 2018). Understanding the extent and the mechanisms by which the effect of LoF variants depends on the genetic background is critical for the development of personalized medicine.

While there are many known examples of background dependencies on LoF mutations, few comprehensive studies have addressed this phenomenon. Studies in *S. cerevisiae* have shown that 5% of essential genes are dispensable between two closely related strains (Dowell *et al*, 2010). In addition, the deletion of 7 chromatin-associated genes was shown to have quantitative differences in growth for 10 different conditions across 2 *S. cerevisiae* genetic backgrounds (Mullis *et al*, 2018). The genetic underpinning of these differences was mapped using 1,411 wild-type and mutant yeast cross progeny revealing a large number of underlying genetic interactions (Mullis *et al*, 2018). In addition to these yeast studies, a systematic RNAi experiment in *C. elegans* showed that 20% of the 1,400 genes tested had different mutant phenotypes across two backgrounds and natural variation in gene expression accounted for some of the observed differences (Vu *et al*, 2015). Recently, gene deletion libraries were generated for 3 other backgrounds of *S. cerevisiae* (Busby *et al*, 2019) other than the original reference laboratory strain library (Winzeler *et al*, 1999). Growth measurements of these knockout libraries in the presence of statin identified strong differences in gene deletion phenotypes across the four genetic backgrounds. The availability of these libraries now allows for the systematic genome-wide study of the impact of the genetic background on gene deletion phenotypes.

1 European Molecular Biology Laboratory, European Bioinformatics Institute, Wellcome Trust Genome Campus, Hinxton, Cambridge, UK
2 European Molecular Biology Laboratory, Genome Biology Unit, Heidelberg, Germany
*Corresponding author. Tel: +49 6221 387 8156; E-mail: typas@embl.de
**Corresponding author. Tel: +44 (0) 1223 494 610; E-mail: pbeltrao@ebi.ac.uk
†These authors contributed equally to this work

Here, we have measured the growth for 4 *S. cerevisiae* deletion collections in a panel of 38 perturbations. The growth of the gene deletion strains shows a large variation across the genetic backgrounds with an average of 18.5% gene–condition associations changing in each strain across all pairwise comparisons. Genes with the largest number of strain-dependent changes of growth had above-average number of genetic and physical interactions suggestive of a role of genetic interactions in these changes. Conditions eliciting variable growth rates among the wild-type strains tended to have the largest number of strain-specific variation in gene deletion phenotypes. Finally, we have measured the growth profiles of a panel of 1,006 *S. cerevisiae* natural isolates (Peter *et al*, 2018) across the same conditions, identifying several variants associated with differences in growth that we linked to causal genes through the gene deletion analysis.

## Results

### Gene deletion growth measurements for 38 conditions in 4 *S. cerevisiae* genetic backgrounds

We measured growth for 17,186 total gene knockouts in 4 *S. cerevisiae* genetic backgrounds (S288C, UWOPS87-2421, Y55, YPS606) with 3,786 gene deletions measured in all backgrounds. The four strains used are genetically diverse with an average of 5.4 to 5.9 SNPs/kb relative to the laboratory "reference" strain S288C (Winzeler *et al*, 1999; Busby *et al*, 2019; Fig 1A). The deletions were arrayed as colonies in a 1,536 agar plate format and were robotically pinned onto agar plates containing the 38 different conditions (Materials and Methods; Fig 1B). Colony size at the endpoint was used as a proxy for fitness, and deviation from the expected growth was calculated, taking into account the replicate measurements, using the S-score (Collins *et al*, 2006; Kapitzky *et al*, 2010; Nichols *et al*, 2011; Fig 1B). The expected growth model assumes that the fitness of a gene deletion in a given condition should be the product of fitness of the independent perturbations (i.e. gene deletion and stress condition). Positive and negative S-scores indicate gene deletions that confer resistance and sensitivity to a given condition, respectively. The list and description of the 38 conditions is available in Table EV1. These include environmental stresses (e.g. heat, high osmolarity, DNA damage), drugs (e.g. caspofungin, clozapine), metabolic conditions (e.g. amino acid starvation) or combinations of stressors.

In total, we measured 876,956 gene–condition S-scores representing the measurements of resistance or sensitivity of each gene knockout in each condition for the four genetic backgrounds (provided as Table EV2). A statistically significant resistance or sensitivity to a given condition is defined as a growth phenotype. The assay is highly reproducible as measured by the correlation of the S-scores using either 13 conditions that were replicated in two batches (Fig 1C, Pearson's $r = 0.744$, $P < 1E-50$) or 2,293 genes that were spotted as replicates on the plates at different locations (Fig 1D, Pearson's $r = 0.811$, $P < 1E-50$). The correlation of S-scores for pairs of gene recapitulates known functional relationships between genes (Fig EV1), further confirming the high quality of the screening data. We observed large differences in the profile of gene deletion growth measurements for the four different strains (Fig 1E) that can be quantified taking into account the high reproducibility of the assay.

### Quantification of genetic background differences of deletion phenotypes

The gene deletion S-scores for the 38 conditions defines the growth profile of the loss of a given gene. If the growth differences of deleting a gene were the same regardless of the genetic background, these quantitative growth profiles would be highly correlated when comparing the S-scores of the same KO in two different strains. We correlated the S-scores of the same gene deletion for pairs of strains as a measure of similarity of their growth profiles and plotted the distribution of correlations for all genes in Fig 2A. On average, the similarity of S-scores of the same gene knockout in different strains is only marginally higher than the observed for the correlation of scores for random pairs of genes (Fig 2A). The lack of correlation could be explained by the large fraction of KOs with no strong response across the conditions screened, resulting in differences in quantitative scores dominated by technical variability. In line with this, the similarity of S-scores across strains increases for gene knockouts having larger numbers of significant deletion phenotypes (Fig 2A). However, even for gene deletions with many phenotypes the correlation across strains remains low, when compared with biological replicates within the same strain.

In order to identify statistically significant differences of growth in each condition, we used an empirical null model that takes into account the variance of the assay and the mean dependence of the variance for the S-score (Bandyopadhyay *et al*, 2010; Materials and Methods; Fig EV2). For each pair of strains, we identify the gene deletion phenotypes that were significantly shared (Fig 2B, black) or exclusive (Fig 2B, red) to each genetic background (Materials and Methods). We then used only the significantly shared/exclusive phenotypes, excluding other phenotypes (Fig 2B, grey), to calculate the fraction of shared/exclusive phenotypes for each pair of strains. We performed all pairwise comparisons for each strain. We then calculated the average fraction of shared phenotypes that a strain has with each of the other three strains (Fig 2C), which ranged from 58% for S288C to 84% for Y55. This fraction drops further for phenotypes significantly conserved across more strains with 22–51% observed in three strains and 9–24% of gene deletion phenotypes significantly conserved in all four backgrounds (Fig 2C). These highly conserved phenotypes include very central genes relevant for the corresponding responses such as sensitivity to osmotic stress (*hog1Δ*), drug efflux (*pdr5Δ*) and amino acid biosynthesis (*adeΔ*, *metΔ*, *serΔ*, *trpΔ*), among others (Table EV3). Only a small fraction (3–5%) of the phenotypes that are not conserved between pairs of strains show a reversal in sign whereby the deletion causes resistance to the stress in one background but increased sensitivity in another background (Table EV4). We observed the strongest reversal for *met5Δ* exposed to amino acid starvation, which has a strong sick phenotype in YPS and UWOP, but shows increased resistance when knocked out in S288C. Since the S288C KO library is based on the BY4741 *met17Δ* (Cherry *et al*, 2012) strain, a hypothetical explanation for this phenotype reversal could be a positive genetic interaction between MET5 and MET17. We observed few changes in these proportions when varying the significance threshold for calling phenotypes, indicating the robustness of these trends (Fig EV3).

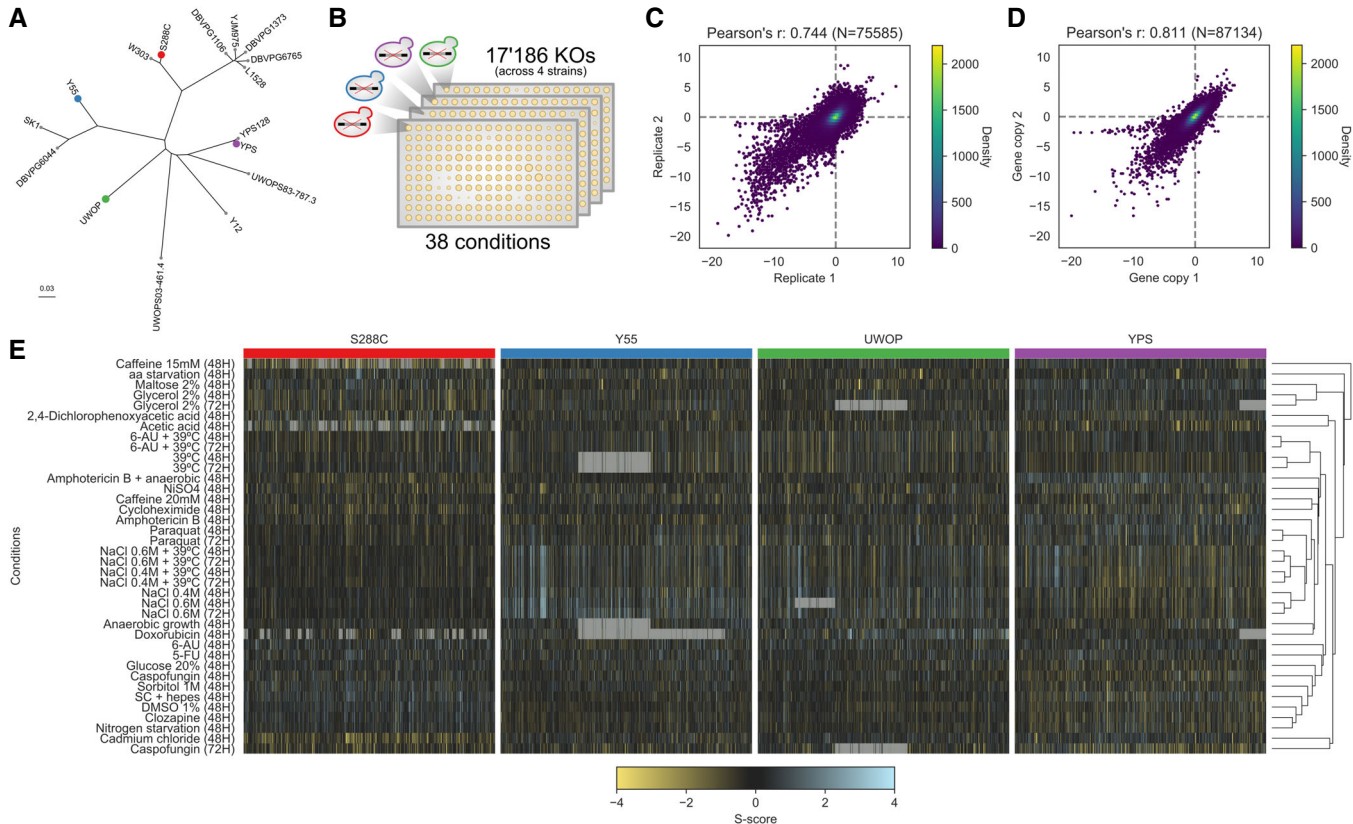

**Figure 1. Chemical genomic screen across four *Saccharomyces cerevisiae* strains.**

A Core genome phylogeny of part of the *Saccharomyces* Genome Resequencing Project (SGRP) yeast isolates; coloured dots indicate the four strains whose KO library was screened in this study.

B Schematic of the chemical genomic screen; each strain's KO library was robotically plated on 1,536 solid agar plates, and each KO colony size was used as a proxy for fitness in each condition.

C Reproducibility of the S-scores using the two batches used in the screening.

D Reproducibility of the S-scores using genes having multiple independent colonies plated in the screening.

E Clustered heatmap of the whole chemical genomic screen; each subsection belongs to an individual strain's KO library. Grey cells indicate missing values.

To ascertain the degree of errors in the quantification of changes in gene deletion phenotypes, we performed two independent assessments of the false-positive rate. For the first test, we analysed as biological replicates 2,326 gene deletions that are spotted in duplicate on all of the four genetic backgrounds. We determined the gene–condition S-scores for each replicate separately and determined the fraction of growth phenotypes that were different when comparing the replicates (Materials and Methods). Any difference in the phenotypes determined for the replicates can be considered an error, giving us an estimated error rate. At a *q*-value below 0.01 (per strain), we detected 2.6, 3.5, 4.5 and 0% false positives for S288C, UWOP, Y55 and YPS, respectively. Additionally, to account for errors in the generation of deletion strains or accumulation of other issues associated with large libraries (e.g. secondary or compensatory mutations, wrong mutations, cross-contamination), we created 16 new deletion strains (Materials and Methods) in each of the four backgrounds. We then performed a smaller scale screen on 10 conditions and determined gene–condition S-scores (Table EV5). The average correlation coefficient of the S-scores between the new replicated KOs and the library KOs was typically higher than 0.6 for

all strains. The exception was the S288C laboratory strain, which tended to have lower correlations (Fig EV4). We used the same procedure to determine the error rate in determining a change in phenotypes for the new replicated KOs and found 17.6, 1, 2.7 and 1.59% for S288C, UWOP, Y55 and YPS, respectively. The rate of error was below 3% with the exception of S288C, which was close to 18%. This higher rate of error for S288C is likely due to accumulated secondary or compensatory mutations in the S288C KO library, which is consistent with previous reports (Teng *et al*, 2013). It is likely that these genomic differences will tend to incorrectly inflate the phenotype differences observed for S288C when compared to other strains (Fig 2C).

We next focused on the gene knockouts that had the largest number of background-dependent changes in phenotypes. We ranked all gene deletions according to the proportion of changes over all tested conditions (Fig 2D, Table EV6) and observed that the genes that change their deletion phenotypes at least once (*N* = 242) had also a higher number of genetic and physical interactions partners based on data collected in the BioGRID database (Chatr-Aryamontri *et al*, 2017) for the S288C laboratory strain (Fig 2E; genetic

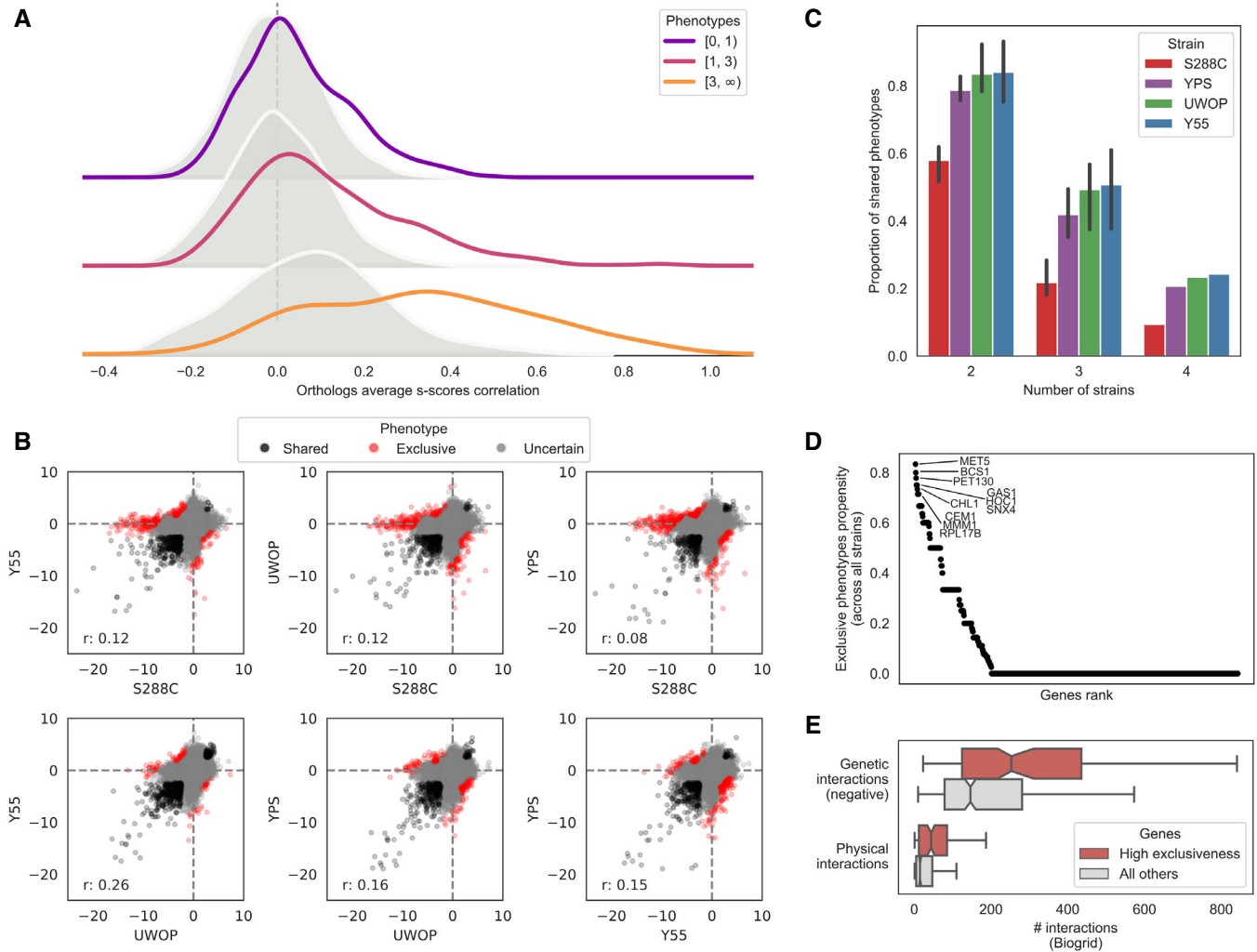

**Figure 2. Systematic assessment of genetic background dependencies of gene deletion phenotypes.**

A   Average S-score Pearson's correlation between the same genes (orthologs, solid line) and random gene pairs (shaded distribution) across all the 38 conditions and four strains. Genes are stratified by the number of conditions in which they show a significant phenotype across the four strains.

B   S-score scatterplots for each pairwise strain comparison, highlighting conserved phenotypes (black points), significant changes (red points) and gene–condition relationships for which no call can reliably be made (grey points). "*r*", Pearson's *r* value.

C   Fraction of deletion phenotypes in each strain conserved with other stains in pairwise, three-way and four-way comparisons. Error bars represent standard deviation for all pairwise and three-way comparisons. Only a four-way comparison is possible for each strain so no error bars are represented in these cases.

D   Gene exclusiveness: a measure of each gene's propensity to change its chemical genomic profile across strains. The top 10 genes' names are reported.

E   Genes with high exclusiveness (> 0) tend to have a higher number of negative genetic and physical interactions (as reported in the BioGRID database). The central vertical line indicates the median, the box delimits the lower and upper quartile of the distribution and the whiskers extend to 1.5 times the Inter-Quartile Range (IQR) plus the lower and upper quartile of the distribution, respectively.

interactions: Kolmogorov–Smirnoff test, *P*-value: 1.00E-9, Cohen's *d*: 0.53; physical interactions: *P*-value: 5.19E-11, Cohen's *d*: 0.49). Significant results are observed also when restricting the analysis to systematic studies reporting at least 2,000 interactions (*P*-value and effect sizes of 1.19E-18 and 0.55, and 1.4E-13 and 0.49 for genetic and physical interactions, respectively). This suggests that the degree of interactions of genes, and therefore their degree of functional connections, is correlated with the probability that a deletion phenotype will depend on the genetic background. It has been shown previously that the number of gene deletion phenotypes of a given gene correlates with its number of genetic interactions

(Costanzo *et al*, 2010). To determine whether our observation depends indirectly on this relationship, we repeated the analysis comparing groups of genes with similar number of growth phenotypes but different numbers of strain-specific phenotypes. This had no impact on the significance of the observed relationship between the number of physical/genetic interactions and number of changes in gene deletion phenotypes.

Genes with many changes in gene deletion phenotypes were enriched in GO terms related to endosomal transport (13 genes over 242, *q*-value = 0.006), mitochondrion organization (23 genes over 242, *q*-value = 0.007), cellular respiration (11 genes over 242,

$q$-value = 0.007) and cell wall organization or biogenesis (20 genes over 242, $q$-value = 0.009; Table EV7). These genes were not more likely than others to be highly conserved ($P$ = 0.065), have a higher fraction of disordered regions ($P$ = 0.49) or show expression changes in optimal growth conditions across the 4 yeast strains (Fisher's exact test $P$ > 0.05; Table EV8).

**Condition-specific wild-type growth differences contribute to variance in gene deletion phenotypes**

For each condition, we counted the number of changes in deletion phenotypes observed across genetic backgrounds (Figs 3A and EV5). We compared these changes with the average growth rate differences of the 4 wild-type (WT) strains (i.e. no knockout) in the same conditions (Figs 3B and EV5). We noticed that some conditions having large number of changes corresponded to cases where a significant growth difference was observed for the WT strains. For example, S288C grows poorly under maltose relative to the other strains (Chow *et al*, 1989) (Fig 3B) and also showed the largest number of exclusive gene deletion phenotypes (Fig 3A), the same was observed for caffeine. In contrast, in the high-salt (osmotic shock) conditions, S288C had the highest WT growth and the smallest number of phenotype differences. While this was not the case for all conditions, there is a significant trend where a slower WT strain growth in a condition is associated with a larger number of strain-specific knockout phenotypes (Fig 3C, Pearson's $r$ = −0.21, $P$ = 0.02).

We analysed in more detail some conditions with large changes in phenotypes. For high salt, that elicits an osmotic stress, deleting the two central kinases of the osmotic shock pathway (HOG1 and PBS2) generally impaired growth in all backgrounds, as expected. This pathway has two upstream branches converging on PBS2/ HOG1 (Brewster *et al*, 1993; Hohmann, 2009). These branches can be redundant and thus show few phenotypes under osmotic stress. However, the STE50 deletion shows striking differences causing increased sensitivity in YPS background, resistance in the Y55 and no phenotype in S288C and UWOP. Similar to STE50, we identified 10 more genes with strain-specific phenotypes in high-salt condition (Fig 3D). Some of these genes are related to osmosensing (STE50, RVS161), ER function (DSC2, GEA1) and metabolism (MRS4, TRP3).

For growth under maltose, the two strains with the best WT growth (UWOP and Y55) also had strong growth defects when maltose-induced genes, present in two clusters, were deleted (Fig 3E). It is known that S288C does not grow effectively in maltose due to inactivation of the maltose activator proteins in the MAL loci (Chow *et al*, 1989). It is therefore expected that deleting MAL genes (i.e. maltose-induced genes) causes no further decrease in growth in maltose-negative strains such as S288C but has a strong impact on MAL-positive strains such as Y55 and UWOP. Similar to S288C, the YPS strain appears to also be a maltose-negative strain. The poor growth under maltose then creates additional vulnerabilities to the cell, rendering essential a large number of genes involved in nonfermentable growth under maltose for S288C (Fig 3E). Interestingly, the same set of genes are required for S288C to grow in glycerol, suggesting that S288C should grow poorly under glycerol, although this did not translate into a strong growth defect of the WT under this condition (Fig 3B).

**Quantitative trait analysis of condition-specific growth differences in a panel of 1,006 *S. cerevisiae* strains**

To test whether the gene deletion phenotypes in different genetic backgrounds could be used to better understand the impact of natural variation on yeast growth under the same conditions, we performed a GWAS across 47 conditions using a panel of 925 *S. cerevisiae* natural isolates. These 925 strains are the fraction of the tested 1,006 strains (Peter *et al*, 2018) with available genotype data. Growth measurements, fitness measurements and phenotype calculations were performed as for the deletion libraries (Materials and Methods). The S-score measurements used (Figs 4A and EV6) represent, as above, condition-specific growth measurements for each strain where a genetic background can specifically affect growth under a given condition. Using the genomic variants, we identified a total of 151,673 common single nucleotide polymorphisms (SNPs, minor allele frequency > 5%). In addition, we predicted the impact of missense variants in each coding region and calculated a probability of loss of function (LoF) for each gene in each strain (Jelier *et al*, 2011; Galardini *et al*, 2017); this could be regarded as a gene disruption or gene burden score. We then performed a GWAS for each condition using the common SNPs, the gene burden score, the gene copy-number variation (CNV) and the presence/absence patterns of genes as predictors (Materials and Methods). In total, we found 579 significant associations (association $P$ < 1E-6), with the largest number of associations observed for growth under amphotericin B and caffeine (Fig 4B), both known to have an impact on the cell wall. Both conditions are also unlikely to be present in the natural environment, and therefore, genetic variants causing growth defects under these conditions are less likely to be selected against. Common SNPs had the highest number of significant associations (365), followed by gene presence/absence (159), gene burden score (29) and CNVs (26). It is not unexpected that SNPs result in the largest number of significant associations since these also constitute the most frequent type of genetic change. Relative to all tested associations, the significant associations represented 0.005% for SNPs, 0.805% for gene presence/ absence, 0.047% for gene burden and 0.06% for CNVs. Not surprisingly, the SNPs had the lowest frequency of significant associations since the other genetic changes are more likely to result in a phenotype difference with a large effect.

For each condition, we obtained a list of genes associated with growth differences from the gene deletion analysis and crossed it with the variants, and their linked genes, associated with growth differences across the 925 yeast strains. Unexpectedly, we found no significant enrichment between the gene–condition associations obtained from the GWAS analysis and the gene–condition associations found in the gene deletion experiments (Fisher's exact test, $P$ > 0.05). Despite the lack of overall enrichment, several GWAS associations can be validated by the gene deletion information (Fig 4C–G; Table EV9). For example, 141 strains had a high gene burden score in the PDR5 locus, which had a significant association with growth in the presence of cycloheximide. Deletion of this ABC transporter is known to cause multidrug sensitivity and showed cycloheximide-dependent deletion phenotypes in all 4 backgrounds (Fig 4D). The presence of two SNPs in two other transporters, the cadmium-transporting P-type ATPase (PCA1) and a membrane $Na^+$/Pi cotransporter (PHO89), was linked to growth under cadmium chloride and had also significant gene deletion phenotypes

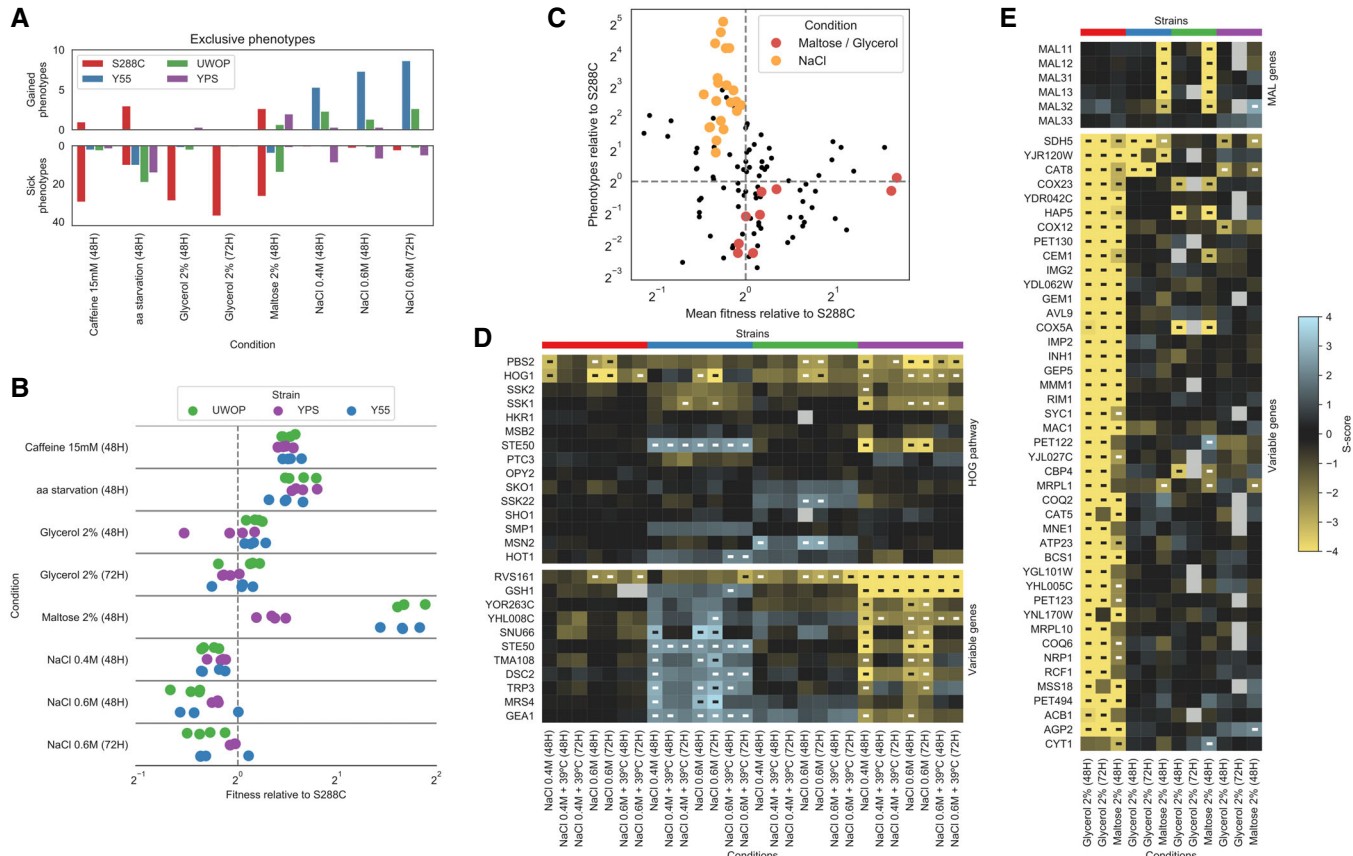

**Figure 3. The fitness of the WT strain background is linked to the number of differences in KO phenotypes for a given condition.**

A Barplots reporting the average number of gain and sick phenotypes that are specific to each strain across all pairwise comparisons.

B Wild-type fitness of each strain relative to S288C across the same conditions as in panel A; each dot represents a specific replicate where colony sizes were measured.

C Relationship between the wild-type fitness relative to S288C and the number of conditionally essential genes relative to S288C; each dot represents a strain–condition replicate as in panel B. The conditions maltose, glycerol and NaCl are highlighted.

D Changes in gene deletion phenotypes for growth on osmotic stress conditions. The top heatmap contains genes belonging to the HOG pathway, while the bottom one contains those genes whose growth phenotypes vary the most between Y55 and YPS. Significant growth phenotypes are marked with "-".

E Changes in deletion phenotypes for growth on glycerol and maltose. The top heatmap contains the MAL genes, while the bottom one contains those genes whose growth phenotypes vary the most between S288C and the other three strains. Significant growth phenotypes are marked with "-".

in at least 3 of the strains (Fig 4E). A SNP close to BRP1 showed an association with growth under high-salt stress, which is supported by BRP1 deletion phenotypes in two strains. However, several other cases had less support. For example, the absence of the ADH1 gene in 450 strains showed an association with anaerobic growth in the presence of amphotericin B. Yet, deleting this gene results in a strong phenotype for in this condition only in the Y55 strain. We found a total of 22 gene–condition associations overlapping between the GWAS and KO analysis with most overlaps observed with gene deletion phenotypes exclusive to a single genetic background.

## Discussion

Our results show that the genetic background has a strong impact on gene deletion phenotypes in *S. cerevisiae*. The fraction of significant differences across two individuals (18–40% and 18.5% on average) is similar to the fraction of changes observed for RNAi

phenotypes for two strains of *C. elegans* 20% (Vu *et al*, 2015). These results also corroborate a high degree of changes in gene deletion phenotypes observed in a study of yeast chromatin-related genes (Mullis *et al*, 2018) and with the initial characterization of the gene deletion library used in this study (Busby *et al*, 2019). The genome-wide nature of our study argues against the possibility that the high degree of changes observed in previous studies could be due to a specific selection of genes. The study of 4 genetic backgrounds also allows us to quantify the degree of shared phenotypes across additional strains. We see that this fraction decreases further with < 25% of significant phenotypes being shared across all four strains. Analysis of additional backgrounds would be needed to fully access the fraction of gene deletion phenotypes that are independent of the genetic background. As with other analyses of large-scale gene deletion libraries, our estimation of background dependence of gene deletion phenotypes may be overestimated due to the errors in the libraries. These errors were found to be typically low (< 5% of error) with the exception of error estimates for newly made KOs in

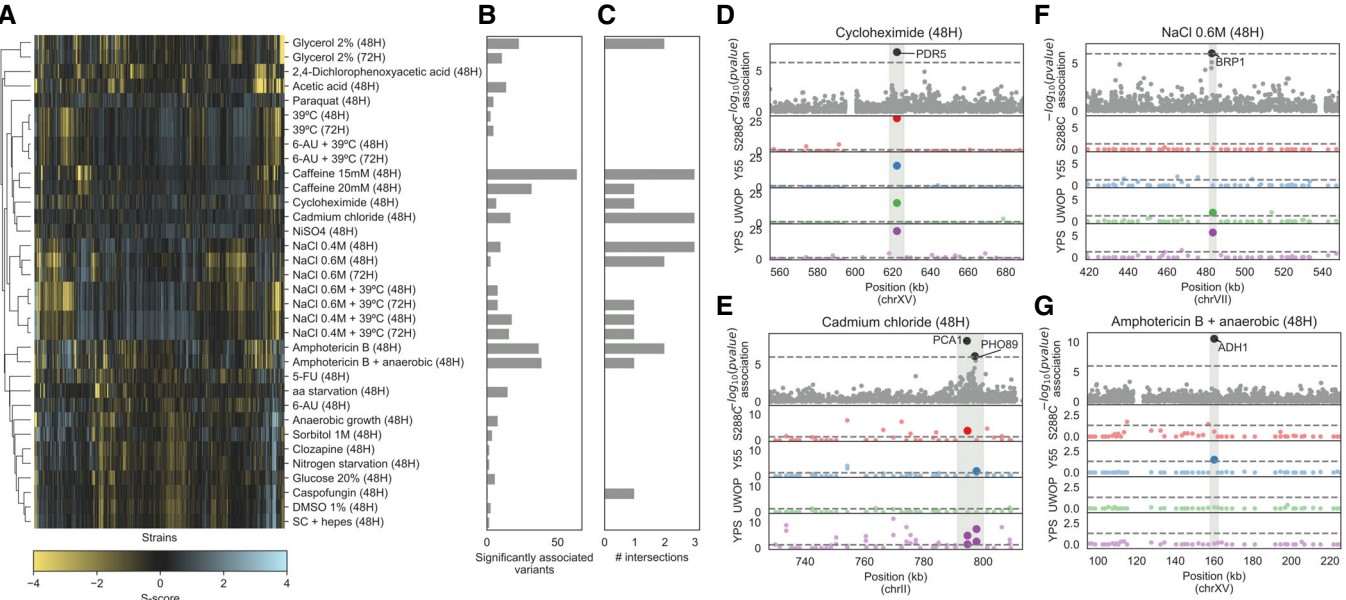

**Figure 4. Genes linked to growth phenotypes via GWAS analysis in 925 natural yeast isolates.**

A  S-score heatmap of the yeast natural isolates across 34 conditions that were also used in the KO screening.

B  Number of variants significantly associated ($P < 1E$-6) with phenotypic variation in each growth condition.

C  Number of associated variants that overlap (i.e. are in a 3-kbp window) with a conditionally essential gene in the same condition, in any of the four KO libraries.

D–G  Manhattan plots showing examples of overlaps between associated variants and the KO screening. The top plot shows the associations between variants and growth in the natural isolates as a function of the $-\log_{10}$ of the association *P*-value, while the four bottom plots show the strength of the KO phenotypes across the four yeast strains, as a function of the $-\log_{10}$ of the corrected S-score *P*-value. Sections shaded in grey indicate the overlap between associations and KO data. Position in the yeast chromosome is reported in kilobase units.

the S288C strain. Based on comparisons with newly made KOs, we found that 18% of gene deletion phenotype changes determined for S288C could be errors potentially due to additional mutations accumulated in the library. If we were to remove 18% of gene deletion phenotype differences that were estimated for comparisons of S288C with other strains, we would still expect to measure on the order of 38% of gene deletions having different phenotypes. Overall, together these studies show very strong evidence that the consequences of the loss of function of a gene can be highly dependent on the genetic background.

The large-scale analysis allowed us to search for general trends associated with the observed differences. Strains having a slower WT growth in a given condition also tended to have a larger number of gene deletion phenotypes in those conditions, suggesting that in such conditions, the poor growing strains have more modes of failure and are impacted by a larger number of gene deletions. Growth in maltose serves as a good example of how existing genetic variation can interact with LoF mutations. The S288C strain has genetic variants that render it unable to grow well in maltose and therefore, in this condition, becomes reliant on genes required for nonfermentable growth. It remains a challenge to find similar justifications for how the genetic background interacts with the gene deletions for other conditions, but those identified here could be further studied using a segregant analysis as previously done by Mullis and colleagues (Mullis *et al*, 2018). Analysing a larger number of such genetic interactions will be fundamental for deriving general principles for how gene deletion phenotypes change across genetic backgrounds.

Some genes had a higher proportion of changes in their deletion phenotypes. These tended to also have an above-average number of genetic and physical interactions. The interaction assays used as the basis for this analysis have been conducted in the S288C background strain, but they nevertheless likely reflect the degree of functional interactions of each gene. Compared to other genes, those that have many genetic and physical interactions have been previously shown to be multifunctional and more important to the cell (Yu *et al*, 2008; Costanzo *et al*, 2010). One interpretation of the results would be that genes that are involved in multiple processes are more likely to have also larger number of changes in deletion phenotypes since there will be many ways by which the genetic background difference may interact with the LoF of these genes. Of the four strains, the reference strain (S288C) stands out as the one where specific gene–condition associations are the most abundant when compared to the other three strains: 58% versus ~80%. This observation combined with other idiosyncrasies specific to this strain (Mortimer & Johnston, 1986; Winston *et al*, 1995; Brachmann *et al*, 1998), such as growth in the presence of maltose, indicates that observations made on a domesticated individual might not necessarily reflect natural populations. In addition, as discussed previously, this laboratory strain gene deletion collection is also more likely to contain secondary and compensatory mutations that may inflate the degree of measured strain-specific deletion phenotypes.

Lastly, we performed a GWAS analysis using 925 strains for the same conditions and using the same experimental set-up. We were

expecting that variants associated with differential growth in a given condition were linked to genes whose deletion resulted in phenotypes in the same condition. Overall, we found no such enrichment, which suggests that using functional genomic assays to study gene–trait associations derived from GWAS may be challenging. This degree of overlap could be limited due to several technical reasons. Natural isolates are likely to have few LoF variants, in particular associated with conditions that are experienced in the environment, and therefore, the genetic perturbations are less likely to mirror the deletion phenotypes. For SNP-based associations in particular, linking the SNP to the causal gene is difficult and could be driven also by long-range genomic interactions. The total number of strains used is likely to be also limiting, since there are typically very few strains showing strong growth differences across any specific condition. Larger number of strains or segregant analysis (Bloom *et al*, 2013; Cubillos *et al*, 2013) could be used in the future to further study the relationship between natural variants and deletion phenotypes. Despite an overall lack of enrichment, our results suggest that in some cases, the interpretation of the impact of genetic variants is possible. Yet, it is unlikely that this would be comprehensive using the gene deletion information available for a single genetic background.

In summary, the results presented here, together with related literature, indicate that gene deletion phenotypes of individuals are strongly dependent on the genetic background. The extent of these dependencies in humans remains to be fully explored but may have important consequences for human genetics.

# Materials and Methods

### Strains used

Mata haploid KO libraries in the genetic backgrounds of S288C (Winzeler *et al*, 1999; 4,889 KOs), UWOPS87-2421 (abbreviated as UWOP, 4,014 KOs), Y55 (4,190 KOs) and YPS606 (Busby *et al*, 2019; abbreviated as YPS, 4,093 KOs) were used to assess whether different genetic backgrounds have an effect on gene deletion phenotypes. These libraries were maintained on YPD+G418 prior to screening in 384 colony format. The 1006 natural isolate strain collection (Peter *et al*, 2018), a kind gift from Gianni Liti (Peter *et al*, 2018), was maintained on YPD in 384 colony format prior to screening.

### Chemical genomic analysis

Growth of KO libraries and 925 strain collection were evaluated on concentrations of chemical and environmental stress conditions (Table EV1) that inhibit the growth of S288C by approximately 40%. The libraries were maintained and pinned with a Singer RoTor in 1,536 colony format. Synthetic complete (Kaiser *et al*, 1994) media were used with or without the stress condition, incubated at 30°C (unless temperature was a stress) for 48 h or 72 h, and imaged using a SPIMAGER (S&P Robotics) equipped with a Canon Rebel T3i digital camera.

The growth measurements were performed in two separate batches with overlapping sets of conditions to judge for variation of the method. In each batch, four biological replicates were collected for each condition tested, and analysis was carried out with a minimum of three replicates. The first batch of the chemical genomic screening was carried out with the S288C deletion collection and used the following conditions: anaerobic, amphotericin B, nystatin, DMSO, 2,4,D, glycerol, maltose, HEPES-buffered medium, caffeine, 6-AU, paraquat, 39°C and sorbitol. Batch 2 was carried out with the deletion collections in the four genetic backgrounds (S288C, Y55, YPS606 and UWOPS87-2421) and contained all (13) of the conditions from batch 1 plus: 5-FU, doxorubicin, cadmium chloride, caspofungin, clozapine, Nickel sulphate, clioquinol, high glucose (20%), minimal medium, nitrogen starvation medium, cycloheximide, and sodium chloride 0.4 and 0.6 M, and the duel stress conditions: sodium chloride 0.4 and 0.6 M plus 39°C, 6-AU plus 39°C and amphotericin B plus anaerobic growth. The two batches were carried out as separate experiments.

### Chemical genomic data analysis

Raw plate images were cropped using ImageMagick to exclude the plate plastic borders. Raw colony sizes were extracted from the cropped images using gitter (Wagih & Parts, 2014), v1.1.1, using the "autorotate" and "noise removal" features on. Poor-quality plates were flagged when no colony size could be reported for more than 5% of colonies (poor overall quality) or when no colony size could be reported for more than 90% of a whole row or column (potential grid misalignment); known empty spots in each plate were used to flag incorrect plates. Overall, < 5% of all pictures have been discarded (175/4,221, 4.15%). Conditions with less than three replicates across the two experimental batches were excluded from further processing. Raw colony sizes for the remaining conditions were used as an input for the EMAP algorithm (Collins *et al*, 2006), with default parameters except the minimum colony size which was set to five pixels. The algorithm computes an S-score, which indicates whether the growth of each KO is deviating from the expected growth in each condition taking into account the variability across the four replicates. The raw S-scores were further quantile-normalized in each condition. Significant loss-of-function and gain-of-function phenotypes were highlighted by transforming the S-scores in *z*-scores, given that the S-scores in each condition follow a normal distribution. *P*-values were derived using the survival function of the normal distribution and corrected using an FDR of 5% (false discovery rate). The whole dataset, comprising 876,956 gene–condition interactions, is available in Table EV2.

The overall relative fitness of the three nonreference strains (Y55, UWOP and YPS) against the S288C reference was computed as follows:

$$\varphi_{strain} = \frac{median(S_{strain})}{median(S_{S288C})}$$

where $median(S)$ is the median-normalized colony size. The normalized colony size is computed in each plate by first applying a surface correction step, followed by a border correction step. The surface correction is applied to reduce the impact of spatial abnormalities on colony sizes; in short, the second-degree polynomials of the row and column indices are computed in a matrix, which is then qr-factorized. The resulting matrix is used to construct an ordinary least squares linear model between the *Q*-factorized

matrix and the corresponding vector of raw colony sizes. The surface-normalized colony sizes are then computed as follows:

$$S_{surface} = S_{raw} - \hat{S} + \underline{S_{raw}}$$

where $\hat{S}$ is the size prediction from the ordinary least squares model. The surface-corrected colony sizes ($S_{surface}$) are further corrected to take into account the border effect, meaning the difference in size between colonies in the two outermost rows and columns with respect to the rest of the plate. The border correction is computed as follows:

$$S_{border} = \frac{S_{outer} \cdot median(S_{inner})}{median(S_{outer})}$$

where $median(S_{outer})$ is the median size of colonies on the outer border of the plate, and $median(S_{inner})$ is the median size of colonies in the rest of the plate. The overall fitness is computed only in those cases where the plates belonging to the four strains' KO libraries have been screened at the same time, in order to make the comparison robust to changes in experimental conditions. The relative fitness measures are available in Table EV10.

To test whether the four KO libraries are able to recapitulate known gene functional relationships, we tested whether gene pairs belonging to the same functional groups tended to have correlated S-score vectors in each of the four yeast strains. Two functional relationships sets were used: the CYC2008 protein complex set (Pu *et al*, 2009) and KEGG modules belonging to *S. cerevisiae* (Muto *et al*, 2013). The KOs common to all four libraries were selected, and for each strain, only those that had at least one phenotype with corrected *P*-value below 0.01 were used to compute the Pearson correlation of S-scores between each gene pair. The ability of these gene–gene correlations to recapitulate the known functional relationships was assessed by constructing receiver operating characteristic and precision–recall curves, using the known relationships as the true-positive set and 10 random gene pair sets with same number as the true set as true-negative set.

The conservation or similarity of gene deletion phenotypes across the four yeast strains was assessed by computing Pearson's correlation between the S-score profiles across all conditions of the same genes in all pairs of strains and then by computing the average of these values. Genes were stratified by the average number of conditions in which they show either a loss-of-function or a gain-of-function phenotype across the four strains. Random gene pairs were used as background.

**Reproducibility of chemical genomic screens**

The reproducibility of the chemical genomic screen was assessed in two separate ways. The first method assessed the technical reproducibility of the S-scores across the two batches in which the screen was conducted. The raw pictures were divided according to the batch of origin, and the EMAP algorithm (Collins *et al*, 2006) was used to compute a set of S-scores for each batch. For the 13 conditions that were tested in both batches, the S-score correlation was computed. We refer to this analysis as both technical and biological because the inoculates are derived from the same source plate but at very different times (Table EV11). Biological replicability was

assessed using 2,293 KOs that are pinned exactly twice across the library.

**Significant changes in growth phenotypes**

Significant changes in chemical genomic profiles between any two strains were computed following a previously published approach that also used S-scores (Bandyopadhyay *et al*, 2010). The two sets of S-scores computed as part of the batch replicate analysis were used as a null model for the absence of changes in S-scores, as a way to estimate the degree of expected variation observed across different experiments. Since the variance in S-scores is higher at higher absolute S-score values, this has to be taken into account when calling significant differences; a sliding window approach was applied when constructing the null model. Given the two sets of S-scores, the following vectors were computed:

$$N_{sum} = -|S_{batch1} + S_{batch2}|$$

$$N_{sub} = |S_{batch1} - S_{batch2}|$$

where $S_{batch1}$ and $S_{batch2}$ are the S-scores from the replicate batches, respectively. The sliding window was then applied to $N_{sum}$, dividing the vector in 100 slices with at least 20 observations in each one and recording the mean ($\underline{N_{sub}}$) and standard deviation ($\sigma_{sub}$) of $N_{sub}$ for each slice. For each strain pairwise comparison, $N_{sum}$ and $N_{sub}$ were recorded for each matching slice, and the corresponding $\underline{N_{sub}}$ and $\sigma_{sub}$ were extracted from the null distribution using a linear interpolation. A normal distribution with mean $\underline{N_{sub}}$ and standard deviation $\sigma_{sub}$ was then constructed around each slice, and the cumulative distribution of the normal function was used to derive a *P*-value to indicate significant differences. The *P*-value was FDR-corrected, and incoherent differences were assigned a corrected *P*-value of 1—specifically, those cases where both strains have a significant phenotype but corrected $P < 0.01$ (150 comparisons over 875,833) and cases where both strains do not show a significant phenotype but corrected $P < 0.01$ (26 comparisons over 875,833). The full dataset comprising 875,834 comparisons is available in Table EV12. We note that this estimate of variance of S-scores is not a condition.

When looking at the proportion of significant loss-of-function or gain-of-function phenotypes that each strain shares with the other strains, we considered those comparisons where the focal strain had a significant gain-of-function or loss-of-function phenotype and corrected $P < 0.01$ (phenotype not shared) and where both strains had a significant phenotype and corrected $P \geq 0.01$ (shared phenotype); all other comparisons were not considered.

Each gene's propensity to change its conditional essentiality profile across the six strains' pairwise comparisons (*E*, exclusive phenotype propensity) was computed as follows:

$$E = \frac{P_{exclusive}}{(P_{exclusive} + P_{shared})}$$

where $P_{exclusive}$ is the number of loss-of-function or gain-of-function phenotypes that vary significantly (corrected $P < 0.01$), while $P_{shared}$ is the number of loss-of-function or gain-of-function phenotypes in both strains of the comparison that do not vary

significantly (corrected $P \geq 0.01$). A gene whose $E > 0$ was considered with "high exclusiveness" (Table EV6). Variable genes in Fig 3D and E were selected based on a corrected $P$-value cut-off of 1E-4 in at least one comparison.

### GO term enrichment analysis

Gene ontology (GO) annotations were downloaded from the SGD database (Cherry et al, 2012), while the GO slim yeast dataset was downloaded from the gene ontology website (Ashburner et al, 2000; The Gene Ontology Consortium, 2017). GO term enrichments were assessed using goatools (Klopfenstein et al, 2018), v0.8.2, using a FDR-corrected $P$-value threshold of 0.01.

### Transcriptomics analysis

Yeast were grown to an OD of 0.4, and total RNA was extracted using the MasterPure Yeast RNA Purification Kit (Biozym, Epicentre). The samples were quality-tested on the fragment analyser (AATi/Agilent) using the Standard Sensitivity RNA Kit (AATi/Agilent), and 600 ng of total RNA was used for library preparation. The libraries were prepared using the TruSeq Stranded mRNA Kit (Illumina) using a Beckman Fxp liquid handler system. Sequencing was carried out on an Illumina NextSeq 500 in 75 single-end mode.

Raw single-ended Illumina reads were trimmed to remove the TruSeq3 adaptors using trimmomatic (Bolger et al, 2014), v0.36. Trimmed reads were pseudo-aligned to the yeast reference genome transcripts (downloaded from the SGD database; Cherry et al, 2012) using kallisto (Bray et al, 2016), v0.44.0, with the sequence-based bias correction and using an average fragment length of 130 bp (70 bp standard deviation). Differential gene expression analysis between each strain and S288C was performed using DESeq2 (Love et al, 2014), v3.8. Raw reads are available in the GEO database with accession number GSE123118.

### Natural isolate growth assay

Raw plate images were cropped using ImageMagick to exclude the plate plastic borders. Raw colony sizes were extracted from the cropped images using gitter (Wagih & Parts, 2014), v1.1.1, using the "autorotate" and "noise removal" features on. Poor-quality plates were flagged when no colony size could be reported for more than 5% of colonies (poor overall quality) or when no colony size could be reported for more than 90% of a whole row or column (potential grid misalignment). Conditions with less than three replicates were excluded from further processing. Raw colony sizes for the remaining conditions were used as an input for the EMAP algorithm (Collins et al, 2006), with default parameters except the minimum colony size which was set to five pixels. The computed raw S-scores were further analysed as reported in "Chemical genomic data analysis" section.

### Genome-wide association study of the yeast natural isolates

The genetic variants found in the yeast natural isolate collection (SNPs/InDels, CNVs and genes presence/absence patterns) were downloaded from http://1002genomes.u-strasbg.fr/files/ on 4 October 2018. SNPs and InDels were normalized and filtered to retain variants with at least 5% minor allele frequency (common variants),

using bcftools (Li et al, 2009), v1.9. Rare variants (SNPs with minor allele frequency $\leq 5\%$) were included computing their impact on gene function, using the "gene disruption score" described in previous studies (Jelier et al, 2011; Galardini et al, 2017). In short, common nonsynonymous and nonsense variants were kept, together with gene presence/absence patterns; the impact of nonsynonymous variants was predicted using the SIFT (Ng & Henikoff, 2001) and FoldX (Guerois et al, 2002) algorithms, when applicable. The individual predictions were translated to their probability of being neutral ($P_{neutral}$) based on a collection of variants with known impact (Jelier et al, 2011), using the following transformations:

$$P(neutral_{SIFT}) = \frac{1}{\left(1 + e^{-(-1.312\ln(SIFT + 1.598E^{-5}) + 4.104)}\right)}$$

$$P(neutral_{FoldX}) = \frac{1}{\left(1 + e^{-(0.218\,FoldX + 0.074)}\right)}$$

where SIFT and FoldX are the scores of each individual predictor. Nonsense variants were assigned a $P_{neutral}$ value of 0.99 if they appeared in the last 5% of the protein sequence, 0.01 otherwise. The overall likelihood that gene function was affected by common variants ($P(AF)$, or gene disruption score) was computed as follows:

$$P(AF) = 1 - \prod_{i=1}^{k} P_i(neutral)$$

where $k$ is the totality of nonsynonymous and nonsense variants observed in each gene. When both SIFT and FoldX predictions were available, priority was given to SIFT scores; nonsynonymous variants with neither predictions available were assigned the highest observed $P_{neutral}$ value. If a gene was considered as absent, a $P$ ($AF$) value of 0.99 was assigned. The gene disruption score is available in Table EV13. The CNVs, gene presence/absence patterns and digitized gene disruption scores (1 if $P(AF) > 0.9$, 0 otherwise) were encoded in a VCF (variant calling format) file together with the common variants and recoded using PLINK (Purcell et al, 2007), v1.90b4.

A genome-wide association analysis has been carried out to highlight common and rare variants associated with growth variability across the yeast natural isolates, using the LMM (linear mixed model) implemented in limix (preprint: Lippert et al, 2014), v2.0.0a3. Missing values were mean-imputed, and the model was applied to each growth condition independently. The kinship matrix was computed using the strains' phylogenetic tree from the original yeast natural isolate publication (Peter et al, 2018). Variants with association $P < 1E-6$ were considered associated. Intersections between associated variants and genes present in the KO libraries were recorded using a 3-kbp window centred around each gene. Gene annotations were retrieved from the SGD database (Cherry et al, 2012). Enrichments were tested using Fisher's exact test.

### Confirmatory KO screening

Gene knockouts for PBS2, HOG1, STE50, PET150, MAL32, MAL12, MET5, BCS1, CHL1, SLM1, SLM4, AVL9, HOC1, ERG2, ERG5 and RPL12B were constructed de novo using the PCR-mediated gene

Marco Galardini et al *Molecular Systems Biology*

disruption method (Amberg *et al*, 2006) in S288C (BY4741), Y55, YPS606 and UWOPS87-2421. Briefly, the KanMX4 cassette was amplified from pFA6a along with 55-bp homology flanking the gene to be deleted. This was transformed as described in Gietz and Schiestl (2007) into the appropriate strain, and confirmed via PCR. The resulting gene KO mutants were then arrayed in 1,536 format (four technical replicates per KO) along with their original corresponding KO strains and screened against the following stress conditions: 2,4-D, minimal media, acetic acid, caffeine, glycerol, maltose, sodium chloride 0.4 and 0.6 M, nickel and synthetic complete medium using the chemical genomic analysis. Plate pictures were processed into S-scores as described in "Chemical genomic data analysis" section. The confirmatory screening comprising 4,287 gene–condition interactions is available in Table EV5.

To estimate the S-scores reproducibility and the identity of the original Kos, the S-score correlation between the original and the newly made KOs was computed, obtaining an estimate of the identity of the original KOs. A similar analysis was done across different strains (e.g. HOG1 original KO in S288C against HOG1 newly made KO in Y55), obtaining an estimate of the overall changes in gene deletion phenotype.

The error in determining changes in gene deletion phenotypes was estimated using two methods: first, we looked at changes between KOs that are plated multiple times in the original screening and counted the proportion of those changes over all comparisons ($q < 0.01$). Secondly, we computed changes between the original KOs and the newly made ones inside the confirmatory screening, again counting the proportion of significant changes over the total number of comparisons.

### Computer code

Most of the code used to process the data is available at the following URL: https://github.com/mgalardini/2018koyeast. The code is mostly based on the python programming language, using the following libraries: NumPy (Oliphant, 2006), v1.15.2; SciPy (Oliphant, 2007), v1.1.0; pandas (McKinney *et al*, 2010), v0.23.4; scikit-learn (Pedregosa *et al*, 2011), v0.20.0; statsmodels (Seabold & Perktold, 2010), v0.9.0; biopython (Cock *et al*, 2009), v1.71; and DendroPy (Sukumaran & Holder, 2010), v4.4.0. Reproducibility was ensured through the use of snakemake (Köster & Rahmann, 2018), v4.7.0. Data were visualized inside Jupyter notebooks using Jupyter (Kluyver *et al*, 2016), v4.4.0, and using the Matplotlib (Hunter, 2007) and Seaborn (Waskom *et al*, 2018) plotting libraries, versions 3.0.0 and 0.9.0, respectively.

## Data availability

The datasets and computer code produced in this study are available as described below:

- All the screening data presented in the manuscript are available in Tables EV2–EV4, as well as online at https://github.com/mgalardini/2018koyeast.
- Transcriptomics data: Gene Expression Omnibus GSE123118 (https://www.ncbi.nlm.nih.gov/geo/query/acc.cgi?acc = GSE123118)

- Code used to process the data: GitHub (https://github.com/mgalardini/2018koyeast)

**Expanded View** for this article is available online.

## Acknowledgements

We thank Ferris Jung and Nayara Trevisan Doimo de Azevedo for the RNAseq work; Lars Steinmetz for the S288C deletion collection; Paul Atkinson for the Y55, YPS606 and UWOPS87-2421 deletion collections; and Leopold Parts, Gianni Liti and Kevin Roy for critical reading of the manuscript.

## Author contributions

PB and AT designed the study. BPB and CV ran the KO screening. MG and ASD analysed the data. BPB ran the natural isolate screening. BPB generated the new mutants and screened them. PB, AT, MG and BPB wrote the manuscript. All authors read and approved the final version.

## Conflict of interest

The authors declare that they have no conflict of interest.

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
