## [Review Process File · Molecular Systems Biology]

The impact of the genetic background on gene deletion phenotypes in *Saccharomyces cerevisiae*

Marco Galardini, Bede P. Busby, Cristina Viteiz, Alistair S. Dunham, Athanasios Typas and Pedro Beltrao.

Review timeline:	Submission date:	17 th January 2019
	Editorial Decision:	14 th February 2019
	Revision received:	7 th October 2019
	Editorial Decision:	7 th November 2019
	Revision received:	13 th November 2019
	Accepted:	14 th November 2019

Editor: Maria Polychronidou

Transaction Report:

1st Editorial Decision

14th February 2019

Thank you again for submitting your work to Molecular Systems Biology. We have now heard back from the three referees who agreed to evaluate your study. As you will see below, the reviewers acknowledge that the study seems interesting and is likely to be broadly relevant. They raise however several concerns, which we would ask you to address in a major revision.

Without repeating all the points listed below, some of the most fundamental issues refer to the need to include further controls and validations in order to better support the conclusions of the study. The reviewers provide constructive suggestions in this regard. Moreover, as the reviewers suggest, the study needs to be better contextualized considering previous studies on the same topic.

All other issues raised by the reviewers need to be satisfactorily addressed. As you may already know, our editorial policy allows in principle a single round of major revision so it is essential to provide responses to the reviewers' comments that are as complete as possible. Please feel free to contact me in case you would like to discuss in further detail any of the issues raised by the reviewers.

REFeree REPORTS

Reviewer #1:

The authors attempt to quantify the extent to which the effects of thousands of gene deletions vary across 4 different strains of budding yeast in >30 growth conditions. The basic conclusion of the study, if correct, is very interesting: that the effects of deleting a gene often change in different genetic backgrounds of the same species. Similar conclusions have been reported before (see below) but the extension to more strains and conditions in this study is potentially interesting. However, the

authors need to better quantify the extent to which the differences are technical artefacts due to variability in the growth assays or errors in the strain collections. At the moment it could be that the authors are overestimating the number of differences. It is also rather unsatisfying that the authors do not dig more into the causes of differences across genetic backgrounds.

[1] Reliability of the data and validity of the conclusions

[a] The reproducibility of the data is colony size data is low ($r^2 \approx 0.5$ by their measurements when the same strains are re-tested) and very little of the data is replicated. This is not unusual for this assay, but it does mean that replication is crucial to such experiments. I think it is therefore essential that the authors properly measure the reliability of their calling of 'differential' phenotypes by re-testing a large number of differential and conserved deletion phenotypes with a decent number of replicates. At the moment we do not have a decent measure of the reliability of the calling of differential and conserved interactions and the authors need to properly estimate the false discovery rates of their calling of 'differential' phenotypes of gene deletion between strains.

 Validation of the deletion strain genotypes

A second likely source of technical errors in the experiment is the fact that deletion libraries notoriously contain clones that are incorrect or mixed. The original *S. cerevisiae* deletion collection, for example, has a substantial error rate for the identity of clones. The authors should experimentally validate the genotypes of a set of the deletions that change effects between strains to quantify the extent to which errors in the libraries underlie the apparent differences in deletion phenotypes between the strains.

[c] Suppressor mutations.

Finally, many strains in the standard deletion collection carry suppressor mutations (for example via aneuploidy) and reconstructing the same deletions reveals a stronger phenotype. To what extent do acquired suppressor mutations in the backgrounds account for the differences in deletion phenotypes between strains?

[2] Relationship to previous work

There are previous analyses of the extent to which deletion phenotypes change across yeast strains: Busby et al (cited) but also Dowell ... Boone Science 2010, which is not cited in this manuscript. To what extent: (1) do the data reproduce between these studies, (2) are the conclusions the same or different. Also (3) what does the current study add in terms of fundamental insights that goes beyond the conclusions of the two previous studies and also the one in *C. elegans*?

The recent paper from Ehrenreich and colleagues (Nat Comm 2018) is also very relevant and should be better summarised in the introduction. The authors of that study mapped the loci underlying differences in the effects of 7 different gene deletions across 10 different conditions, identifying > 1000 genetic interactions.

[3] Data analyses

The authors identify a few genomic features that associate (to some extent) with more variable deletion phenotypes across strains. However these 'explanations' are quite weak and it is rather unsatisfying that they cannot account for more of the variance. In the end this may simply be the case, but there are plenty more features (experimental and sequence-based) that the authors could test.

More minor point: the definition of pleiotropy is rather strange. The authors should look through the yeast genomics literature for the alternative metrics that have been used (Zhang, Myers etc).

[4] "We further illustrate how these changes affect the interpretation of the impact of genetic variants across 925 yeast isolates."

I do not think the authors have actually done this and this sentence in the abstract misrepresents what is shown in the results ('Unexpectedly, we found no significant enrichment between the gene-

condition associations obtained from the QTL analysis and the gene-condition associations found in the gene deletion experiments"). The results presented in the last section of the results are, unless I missed something, rather anecdotal and it is not clear whether the condition-specific deletions do or do not affect the interpretation of the impact of genetic variants. A systematic analysis is needed here.

[5] minor comments/corrections

p 9 'In total we found 579 significant associations..' - what is the FDR or significance threshold ?
Fig 1a. The labels are too small to read.

- Fig 1e. gene clustering + ontology analysis would be useful to see tendencies and make the figure easier to interpret.

- Fig 2a. The classification of phenotypes here seems rather arbitrary

- Fig 2c. What are the error bars? Why there is no error bar for the "4" number of strains?

- Fig 2d. How many of the "exclusive" phenotypes are actually significantly switching between at least two strains? I would highlight them in different color, would be easier to see that these are 3-5% of the cases in addition to the suppl table.

- Fig 4f/g not mentioned in the text.

- Some numbers are written as with or without commas (eg. 1000 vs 1,000)

- typo: "To test whether our the gene deletion..."

Reviewer #2:

The authors examine how the phenotypic effects of gene deletions differ across four budding yeast strains. They examine the majority of genes in the genome, meaning their dataset is comprehensive and capable of producing general insights. They find that a sizable fraction of all gene deletions show phenotypic effects that differ across strain backgrounds. This is an important result. Overall, I found this paper interesting and likely to generate broad interest. Some comments:

1. Definition of incomplete penetrance in abstract is not technically correct. Note, incomplete penetrance refers to when a mutation does or does not show a phenotype. It is a qualitative phenomenon. However, the abstract describes a quantitative phenomenon, more akin to variable expressivity. No one is greatly interested in these semantics, so the simplest solution may be to delete ', a phenomenon known as incomplete penetrance.'
2. In multiple places, wordy, jargon phrases are used, such as 'condition specific growth phenotypes' and 'gene-condition phenotypic interactions.' Simpler, less opaque wording might be possible. This should also make it clearer to the reader what the 876,956 number is describing.
3. Figure 3a: the dots and labels are hard to see. Please make them bigger.
4. The following sentence was very difficult to digest: 'The gene deletion phenotype scores for the 38 conditions were correlated across pairs of strains as a measure of similarity of their phenotypic profiles and plotted as a distribution for all genes in Figure 2A.'
5. I also found the wording here confusing: 'We performed all pairwise comparisons and for each strain we then calculated the average fraction of shared phenotypes with the other 3 strains (Figure 2C), which ranged from 58% for S288C to 84% for Y55. This fraction drops further for phenotypes significantly conserved across more strains with 22% to 51% observed in 3 strains and 9% to 24% of gene-deletion phenotypes significantly conserved in all 4 backgrounds (Figure 2C).' Is the first sentence missing something like 'each of' prior to 'the other 3 strains.'
6. Can some global measure of similarity or correlation in responses across all gene deletions be provided for each pair of strains?
7. Often the word 'phenotype' is used to describe the 'phenotypic effect' of a gene deletion, e.g. p5. It might be good to change this as the phenotypes the authors are measuring are growth in different environments. They then use the growth phenotypes to determine the effects of particular deletions in specific environments and backgrounds.

8. P5, expression data not generated for all conditions, so final sentence on this page merits qualification.
9. Figure 2 legend, should be 'each gene's propensity.'
10. Bottom of p7, 'into a strong growth defect.'
11. Figure 3b legend, it should be noted that maltose, glycerol, and NaCl are highlighted. It took me a moment to figure out what was going on here given the figure and the existing information in the figure legend.
12. Figures 3d and 3e. I could not figure out what the dash/minus symbols meant. Please add to legend.
13. P9, most would not refer to a GWAS study as a 'QTL analysis.' The latter typically refers to a mapping study focused on a known pedigree, such as a controlled cross.
14. P9, it is not clear what the percentages refer to. I would have expected they refer to the proportion of the associations, but that cannot be the case.
15. The discussion is somewhat superficial, mainly just recapping the results of the paper. I wondered if the discussion couldn't be built out a bit, connecting the paper more to the broader literature and problems. Also, the wording at the end of the discussion could be improved because it is wordy, but also somewhat vague. See: 'Despite an overall lack of enrichment, our results suggest that interpretation of the impact of genetic variants using the gene deletion information available for a single genetic background is unlikely to be comprehensive. In summary our results suggest that interpretation of the impact of genetic variants on the phenotypes of individuals would likely need detailed gene-phenotype information in more genetic backgrounds than that of a model individual.'
16. Apostrophes used instead of commas in numbers equal to or greater than 1,000 in some places, such as first section of Methods.

Reviewer #3:

The paper by Galardini and colleagues is a timely analysis bearing on important fundamental questions about how often and why a genetic variant causes a phenotype only some of the time (incomplete penetrance) or to varying quantitative extents (variable expressivity). Two important sources of incomplete penetrance (effects of genetic background and changes in environment) are explored. The papers use deletion collections in four different yeast strains to estimate how many knockout effects are strain-dependent (when considering growth in different environments). They find many strain-dependent knockout effects, especially those that are specific to S288C. This study is interesting and systematically extends earlier work exploring background-dependent knockout effects in yeast and other organisms. However, I do have some concerns about the analysis and presentation, and overall I think the manuscript can be suitable for publication if major revisions are made.

Major:

- The conclusion that 16-42% of deletion phenotypes change between pairs of strains is interesting, but I am somewhat skeptical of the analysis:
- Regarding the null model: When trying to find significant differences between the growth of two knockouts, a null model should take into account the expected variability in both knockout measurements. However, it seems to me that their null model only considers variability in one knockout effect (N_{sub}), and not both. If I am mistaken, this should be more clearly explained. If this is correct, the null model should be changed to include the estimated variance of the effect in both strain (i.e. a pooled variance estimate), otherwise differences will be over-called.
- Related to the above point, how were the comparisons made when one or both of the conditions were not measured in two batches? The authors state that only 12 conditions were measured in two batches, but 38 were measured in total. It may be possible for the authors to have an (imperfect)

variability estimate if something was measured only one batch (e.g. between their internal controls), but I do not see this described anywhere.

-Even though the nominal FDR for differential growth analysis is low, the assumption of the 'null' FDR model is that the variability between two replicates is normally distributed. However, some data artifacts are evident which raise concern about false positives (which can inflate estimates of variable knockout phenotypes). For example, in both Figure 1C and 1D, there are some genes which seem to have a negative S score in Replicate 1/Gene copy 1 but not in the other replicate. As a sanity check, the authors should apply their differential fitness calling method between the scores obtained in Batch 1 and Batch 2, or have some other kind of 'empirical' control for how many false differential knockout effects they expect.

- Re: "Even though a part of the observed changes might be false positives, we are confident that the homogeneity in experimental conditions as well as excluding uncertain cases from the analysis (Methods) helps reducing these cases to a small number." An unsupported statement of the authors' confidence is simply not enough here. This issue needs quantitative evaluation. I am especially concerned about false positives because so many knockout effect comparisons are made (e.g. between strains, between environments) - so even if the false positive rate is low, many comparisons might lead to at least one significant difference just by chance. For example, the rate at which significantly different phenotypes are observed can be estimated for replicate batches. Given an estimate of reproducibility for positive and negative phenotype observations, what fraction of genes would show a phenotype in only 1 of 4, 2 of 4, 3 of 4 or 4 of 4 replicates, even when the ground truth remains the same?

-Analysis of genetic and physical interaction degree relies on BioGrid, and therefore is unsystematic and subject to the well-known ascertainment biases of the literature. This analysis should be limited to systematic datasets within evenly tested search spaces. For example, Costanzo et al Science 2016 within the space of query x array genes used for mapping genetic interactions. Similarly, it could be Yu et al Science 2008 for direct protein interactions (within the space of protein pairs tested) or Gavin et al 2006 for co-complex associations (for all pairs involving a protein that was used as bait). It should be noted that correlations between genetic interaction degree and pleiotropy and between genetic interaction degree and single-mutant fitness were shown previously in Costanzo et al Science 2010. Correlation between direct protein interaction degree and pleiotropy was shown in Yu et al Science 2008.

-One potential source of variability is that the KO was properly made in some strains but not others. Quality control should be performed on a random sample of genes for which variation across strain backgrounds has been seen, in which there is a careful PCR-based analysis to make sure that: 1) the deletion cassette is present; 2) the junctions between flanking genomic region and deletion cassette are as expected; and 3) the target gene is actually absent. The latter is the most critical, as a common scenario when knocking out genes with fitness effects is that the only deletions that can be contained were in cells for which the target locus was duplicated, so that the deletion cassette replaces one copy of the target gene but leaves another copy of the target gene intact.

Minor:

-There are some discrepancies between how the experiment is presented in the main text and how it is described in the methods. Specifically, the methods mention three batches (one of them measuring only S288C in some conditions), but two batches are mentioned elsewhere. Which is correct?

-It is shown that high-exclusiveness genes have a higher negative genetic interaction degree, and higher protein protein interaction degree. However:

-Figure 2E should contain a statistical test

-It is known that genetic interaction degree for a gene correlates with its single-deletion effect (Costanzo et al 2010, 2018). Perhaps a similar correlation exists with PPI degree from Biogrid. Given results in Figure 3A, the background set should be chosen to have the same distribution of single-deletion effects (i.e. same mean and standard deviation) as the high-exclusiveness genes to correct for this.

-In Figure 2C, it is unclear why there are different estimates for number of genes shared amongst 4 strains - only 4 strains were tested, so this should just be a single set of genes (i.e. those which did not show a difference for any comparison). Why are there four bars, and why are the numbers different?

-When performing the Fisher's exact test to calculate significant overlap between associations and knockouts, were the genes that did not have usable variants excluded from the calculation? In general, the overlap may be significant if the authors don't consider genes where finding an association is very unlikely (e.g. not enough natural variability), and I think it is fair to do this

filtering

-All association and enrichment analysis should give not only a measure of significance (p- or q-value) but also an estimate of effect size (e.g., enrichment factor). For negative results, giving the actual P-value is preferable to just saying $P > 0.05$)

Issues with text:

- The authors should be consistent about whether they use 's-score' or 'S-score' (I think the latter is more correct)
- Text in Figure 1A is not legible at the size provided
- Colour legend in Figure 2B is not legible
- Gene names should be italicized
- Deletion notation is incorrect - e.g. *pdr5* Δ (italicized), not Δ PDR5
- Cycloheximide is spelled incorrectly, and should not be capitalized
- PDR5 deletion is generally leads to multidrug sensitivity, not resistance
- In general, paper could use more copyediting as there are several other grammar/spelling/formatting mistakes as well (e.g. "indicating the robustness of this trends", "duel stress conditions", "4'889 KOS")
- The word "numerosity" (the discrete property of being numerous) should be replaced with a continuous or cardinal quantifier, like "size" or "number".

Below follows our point-by-point response to the reviewers' comments. Our responses are in blue.

Reviewer #1:

The authors attempt to quantify the extent to which the effects of thousands of gene deletions vary across 4 different strains of budding yeast in >30 growth conditions. The basic conclusion of the study, if correct, is very interesting: that the effects of deleting a gene often change in different genetic backgrounds of the same species. Similar conclusions have been reported before (see below) but the extension to more strains and conditions in this study is potentially interesting. However, the authors need to better quantify the extent to which the differences are technical artefacts due to variability in the growth assays or errors in the strain collections. At the moment it could be that the authors are overestimating the number of differences. It is also rather unsatisfying that the authors do not dig more into the causes of differences across genetic backgrounds.

We thank the reviewer for the positive remarks. We address below the important remarks on reproducibility of the data in support of the conclusions. We agree with the reviewer that getting deeper insights into the causes of the differences would be ideal. However, we think that the best way forward will be to perform crosses to map the genetic causes of the differences which will require substantial amount of work that we hope to do in the future.

[1] Reliability of the data and validity of the conclusions

[a] The reproducibility of the data is colony size data is low ($r^2 \approx 0.5$ by their measurements when the same strains are re-tested) and very little of the data is replicated. This is not unusual for this assay, but it does mean that replication is crucial to such experiments. I think it is therefore essential that the authors properly measure the reliability of their calling of 'differential' phenotypes by re-testing a large number of differential and conserved deletion phenotypes with a decent number of replicates. At the moment we do not have a decent measure of the reliability of the calling of differential and conserved interactions and the authors need to properly estimate the false discovery rates of their calling of 'differential' phenotypes of gene deletion between strains.

We thank this reviewer for bringing up this point, we fully agree that we need to provide a measurement of the reproducibility of calling a change in gene-condition interactions across strains. As we used an empirical background model, we assumed we would have a low fraction of incorrect calls but this needs to be demonstrated. To do so, we performed two different tests to measure the error rate of calling a change - the first test takes advantage of genes that are spotted in the array in two different positions that can act as biological replicates; and a second and more stringent test where we remade and screened again a number of KOs in the 4 backgrounds.

In the original KO collection as well as in the 3 new KO collections there are 2326 genes that are spotted more than once in the array in different locations across plates. For the original analysis we considered only one position per gene. In the absence of variability we would expect not to call any change in gene-condition interaction for the same gene deletion, in the same strain screened under the same condition. We performed an analysis where we determined the gene-condition score and used the model described in the paper to determine the p-value of a change of interaction for all pairs of genes that are duplicated in the plates. Any significant change that is predicted for these pairs constitutes an empirical false positive. We show below the distribution of $-\log_{10}(q\text{-value})$ for the predicted changes for biological replicates. As can be seen in the figure, most q-value values are near the expected 1 value with very few below 0.01.

For each strain the proportion of comparisons with qvalue below 0.01 (per strain) is : S288C 2.6%, UWOP 3.5%, Y55 4.5%, YPS 0%. The largest false positive rate was found for Y55 and is below 5%. Overall, this analysis suggests that the error rate on determining a change in gene-condition interaction is very low (<5%). We have added this analysis of reproducibility to the manuscript.

To further address this concern and more broadly analyse the reproducibility of the screen we created new knock-out strains in each background. We detail the results from this analysis in the next point.

Validation of the deletion strain genotypes

A second likely source of technical errors in the experiment is the fact that deletion libraries notoriously contain clones that are incorrect or mixed. The original *S. cerevisiae* deletion collection, for example, has a substantial error rate for the identity of clones. The authors should experimentally validate the genotypes of a set of the deletions that change effects between strains to quantify the extent to which errors in the libraries underly the apparent differences in deletion phenotypes between the strains.

The reviewer correctly points out that any large strain collection will have some errors due to incorrect IDs, genome stability issues, acquisition of additional mutations, etc. To measure the potential reproducibility of replicating the mutation process and take into account potential sources of error such as the strain gene ID and suppressor mutations (next point from the reviewer) we made KO strains *de novo* for 16 genes in the 4 backgrounds. These mutants were made by targeting a KO cassette to each gene locus by homologous recombination. We verified that all newly made mutants were at their correct gene locus by PCR and sequencing. These

new mutants were screened together for growth phenotypes under 10 conditions for which we derived gene-condition S-scores.

The average correlation coefficient of S-scores for these replicated KOs with the library KOs was typically higher than $r=0.6$ with the exception of the S288C lab strain (see figure below). This is compatible with the possibility that the S288C library may have accumulated a higher number of genetic changes relative to the other libraries that were made more recently. This would also explain why the fraction of changed interactions is apparently higher when comparing with S288C in our study. The difference between the changes measured for the other strains (~20% changing across a pair of strains) and S288C (~40% changing across a pair of strains) may be accounted for, in part, by accumulated genetic changes in the lab strain library. This is an interesting point in itself.

As we did above, we used the measurements of the gene deletion phenotypes of the newly generated mutations as biological replicates and assessed the error rate of determining a change in a gene-condition interaction. Q-value distribution in new screening between KO collection and newly generated KOs:

The proportions of false positive changes predicted per strain are: S288C 17.6%, UWOP 1%, Y55 2.7%, YPS 1.59%. Overall, even for newly generated mutants the reproducibility of predicting a change in gene-conditions is high with the clear exception of changes called between replicates of S288C. Again, this likely explains why the observed changes measured between S288C and other strains was on average higher than for the other strains. We don't think that errors in correct ID of the knock-out gene is a strong factor in the difference in reproducibility between strains. This is due to the fact that the KO collections in the 3 new genetic backgrounds were generated by backcrossing the S288C collection into the 3 new strain backgrounds. So if there are errors in gene deletion IDs, these will be perfectly mirrored in the new collections. We suspect that the S288C collection may have acquired additional mutations that exacerbate the differences in gene-deletion phenotypes.

We have added these additional experiments to the manuscript and revised the corresponding results and discussion sections.

[c] Suppressor mutations.

Finally, many strains in the standard deletion collection carry suppressor mutations (for example via aneuploidy) and reconstructing the same deletions reveals a stronger phenotype. To what extent do acquired suppressor mutations in the backgrounds account for the differences in deletion phenotypes between strains?

We think the experiments described above also address this concern.

[2] Relationship to previous work

There are previous analyses of the extent to which deletion phenotypes change across yeast strains: Busby et al (cited) but also Dowell ... Boone Science 2010, which is not cited in this manuscript. To what extent: (1) do the data reproduce between these studies, (2) are the conclusions the same or different. Also (3) what does the current study add in terms of fundamental insights that goes beyond the conclusions of the two previous studies and also the one in *C. elegans*?

We actually meant to cite Dowell and colleagues in the introduction but instead cited Ryan ... Boone Science 2012 which uses the same $\Sigma 1278b$ collection. We have corrected this citation now. The work by Dowell and colleagues describes the differences in gene essentiality in two different strains and does not consider different stress conditions. We think our contribution goes beyond other studies by performing the first genome-wide formal assessment of gene-deletion phenotypes across an array of different conditions. While the studies in *C. elegans* and the yeast work done by Ehrenreich and colleagues are fantastic they were not done at the genome-wide scale. We think the genome-wide nature of our study is an important contribution since it argues against the possible bias in selecting a specific set of genes (e.g. chromatin regulation). The largest study to date is the *C. elegans* work cited and we think these are highly complementary towards each other, since the genetic perturbation (RNAi vs KO) and model systems (*C. elegans* vs. *S. cerevisiae*) are different. By studying 4 backgrounds our study also allowed us to determine the degree of shared phenotypes across more than a pair of strains. It goes beyond Busby and colleagues in the number of conditions and the formal quantification of the changes observed across strains. Busby and colleagues did not provide a model for the quantification of changes as we do in our study. When compared to the *C. elegans* study, one major drawback of our study and other studies relying on gene deletion collections are issues with library construction and accumulation of mutations such as suppressors. This is well illustrated by the higher rate of errors described above for the calling changes in gene deletion phenotypes of S288C. More importantly, these studies all corroborate each other with different strengths, and point to a very important and still overlooked high degree of genetic background dependencies of the effects of gene loss of function. We attempted to better describe in the discussion section how this study goes beyond the others mentioned.

The recent paper from Ehrenreich and colleagues (Nat Comm 2018) is also very relevant and should be better summarised in the introduction. The authors of that study mapped the loci

underlying differences in the effects of 7 different gene deletions across 10 different conditions, identifying > 1000 genetic interactions.

This work was already cited in the introduction section. We have expanded the description of what was found and also relate this work to our study in the discussion. We also mention this work in the discussion as the most promising way forward to understand the mechanisms that determine the changes in deletion phenotypes.

[3] Data analyses

The authors identify a few genomic features that associate (to some extent) with more variable deletion phenotypes across strains. However these 'explanations' are quite weak and it is rather unsatisfying that they cannot account for more of the variance. In the end this may simply be the case, but there are plenty more features (experimental and sequence-based) that the authors could test.

We agree with the reviewer that it would have been ideal to gain a deeper understanding of the reasons and mechanisms underlying the changes in deletion phenotypes. We think this will require additional experiments. In the manuscript we have analyzed a few key ideas: genes with background dependent changes in KO phenotypes have a higher number of physical and genetic interactions; they are associated with specific cell biology terms; and they are **not** more likely than others to change their basal gene expression across strains. We have tried a few other potential explanations along with the reviewer's suggestion. Genes with changes in phenotypes across backgrounds are **not** more likely than others to be highly conserved (p-value: 0.065) or have a higher fraction of disordered protein sequences (p-value: 0.49, see figures below).

We think that the way forward will be to start dissecting examples by further experiments where we can perform crosses between strains carrying the gene deletions in order to map the genetic variation that causes the changes in gene deletion phenotypes across backgrounds. This is something that are considering doing but it will take considerable effort and goes beyond the scope of this current manuscript.

We have added these additional results and added to the discussion on how to move forward in the future to gain further mechanistic insights into the mechanisms underlying the changes in gene deletion phenotypes.

More minor point: the definition of pleiotropy is rather strange. The authors should look through the yeast genomics literature for the alternative metrics that have been used (Zhang, Myers etc).

We have revised the manuscript to avoid the term.

[4] "We further illustrate how these changes affect the interpretation of the impact of genetic variants across 925 yeast isolates."

I do not think the authors have actually done this and this sentence in the abstract misrepresents what is shown in the results ("Unexpectedly, we found no significant enrichment between the gene-condition associations obtained from the QTL analysis and the gene-condition associations found in the gene deletion experiments"). The results presented in the last section of the results are, unless I missed something, rather anecdotal and it is not clear whether the condition-specific deletions do or do not affect the interpretation of the impact of genetic variants. A systematic analysis is needed here.

We agree with the reviewer that the abstract sentence is not a good representation of the work and the results we obtained. The systematic result that we observed was a negative result - that there is no significant enrichment between the gene-condition associations obtained from GWAS and the gene-condition associations found in the gene deletion experiments. This would suggest that a GWAS for a given trait will tend to identify loci with genes that do not overlap with those relevant for that same trait based on functional genomic assays. Given the amount of effort and resources that have gone into human GWAS studies over the years, this is a troubling result. One obvious difference is that the gene-deletions are a strong loss of function while the common variation local to the associated genes may cause local changes in gene expression of cis-linked genes. These differences in expression are not necessarily expected to cause a loss of function effect. Additionally the associated genetic variants may not be causing changes to cis-linked genes but instead may have long-range effects in complex ways. In this way the mapping of variants to genes and phenotypes in GWAS is a complex problem that may also underlie part of the difference. We think this negative result is quite important in itself and has important cautionary messages to human genetic studies. There is an increase in efforts to combine GWAS with functional genomic assays for human genetic studies. This work suggests that these efforts are going to be difficult.

Despite the overall lack of enrichment the cases where an overlap is seen are still very informative. Linking GWAS associated SNPs to genes is error prone as multiple nearby genes or even distal genes may be the true causal genes. For that reason, the cases where there is an overlap, the gene deletion phenotypes provide an external validation of the GWAS signal and a

putative mechanistic explanation for the association in the region. This idea underlies many ongoing human genetic initiatives where, for example, CRISPR screens are being done in human cell lines of relevance to study GWAS linked genes.

We have revised the abstract to be more explicit about this negative result and have expanded the discussion section further on the relation between the gene deletion and GWAS.

[5] minor comments/corrections

p 9 'In total we found 579 significant associations..' - what is the FDR or significance threshold ?
We have used an association p-value threshold of $1E-6$ for calling significant associations. We have added this information in the results and methods section.

Fig 1a. The labels are too small to read.

We have increased the label size in Fig 1a.

- Fig 1e. gene clustering + ontology analysis would be useful to see tendencies and make the figure easier to interpret.

Figure 1e is meant to give an overview of the data gathering effort and to provide a first indication of the degree of changes observed across strains. It was not clear exactly what the reviewer meant by this comment.

- Fig 2a. The classification of phenotypes here seems rather arbitrary

We agree that other splits could have been done but we think these are meaningful. They represent genes with no, few and many phenotypes. These splits are simply to show that even when excluding genes with few or no phenotypes we still observe a correlation that is overall weak.

- Fig 2c. What are the error bars? Why there is no error bar for the "4" number of strains?

The gene deletion phenotypes of each strain is compared with the other strains in a 2-way, 3-way and 4-way manner and the error bars represent the variation across the comparisons. There are multiple 2 and 3-way comparisons but only one 4-way comparison. This is why the 4-way comparisons don't have a variation bar on them. We have added a description of what the error bars represent to the figure legends.

- Fig 2d. How many of the "exclusive" phenotypes are actually significantly switching between at least two strains? I would highlight them in different color, would be easier to see that these are 3-5% of the cases in addition to the suppl table.

This could not be done since the switching is dependent on each condition separately, while the measure we show is spread across all conditions.

- Fig 4f/g not mentioned in the text.

- Some numbers are written as with or without commas (eg. 1000 vs 1,000)

- typo: "To test whether our the gene deletion..."

We have revised the text to correct for these issues.

Reviewer #2:

The authors examine how the phenotypic effects of gene deletions differ across four budding yeast strains. They examine the majority of genes in the genome, meaning their dataset is comprehensive and capable of producing general insights. They find that a sizable fraction of all gene deletions show phenotypic effects that differ across strain backgrounds. This is an important result. Overall, I found this paper interesting and likely to generate broad interest. Some comments:

1. Definition of incomplete penetrance in abstract is not technically correct. Note, incomplete penetrance refers to when a mutation does or does not show a phenotype. It is a qualitative phenomenon. However, the abstract describes a quantitative phenomenon, more akin to variable expressivity. No one is greatly interested in these semantics, so the simplest solution may be to delete ', a phenomenon known as incomplete penetrance.'

We have changed the text to remove the connection to the term "incomplete penetrance".

2. In multiple places, wordy, jargon phrases are used, such as 'condition specific growth phenotypes' and 'gene-condition phenotypic interactions.' Simpler, less opaque wording might be possible. This should also make it clearer to the reader what the 876,956 number is describing.

We appreciate the desire for clarity such that a broad audience can follow the text. We have revised these terms in the results section.

3. Figure 3a: the dots and labels are hard to see. Please make them bigger.

We assume that the reviewer here meant to say Figure 3c and we have increased the size of that panel as suggested.

4. The following sentence was very difficult to digest: 'The gene deletion phenotype scores for the 38 conditions were correlated across pairs of strains as a measure of similarity of their phenotypic profiles and plotted as a distribution for all genes in Figure 2A.'

We have changed the text to facilitate the interpretation of these results.

5. I also found the wording here confusing: 'We performed all pairwise comparisons and for each strain we then calculated the average fraction of shared phenotypes with the other 3 strains (Figure 2C), which ranged from 58% for S288C to 84% for Y55. This fraction drops further for phenotypes significantly conserved across more strains with 22% to 51% observed in 3 strains and 9% to 24% of gene-deletion phenotypes significantly conserved in all 4 backgrounds (Figure 2C).' Is the first sentence missing something like 'each of' prior to 'the other 3 strains.'

The reviewer is correct. We have split up and corrected this sentence.

6. Can some global measure of similarity or correlation in responses across all gene deletions be provided for each pair of strains?

The correlation between responses across all gene deletions and for each pair of strains is shown in Figure 2B. We have added the requested correlation values to this figure.

7. Often the word 'phenotype' is used to describe the 'phenotypic effect' of a gene deletion, e.g. p5. It might be good to change this as the phenotypes the authors are measuring are growth in different environments. They then use the growth phenotypes to determine the effects of particular deletions in specific environments and backgrounds.

We have revised the text to be clearer about the usage of the word "phenotype". We say now in page 4: "A statistically significant resistance or sensitivity to a given condition can be defined as a condition specific growth phenotype". We have replaced some instances of the term "phenotypes" with "growth measurements" or "S-scores" for clarity.

8. P5, expression data not generated for all conditions, so final sentence on this page merits qualification.

We have revised this text section to make this more explicit.

9. Figure 2 legend, should be 'each gene's propensity.'

We have clarified this in the figure legend.

10. Bottom of p7, 'into a strong growth defect.'

Change made in the text.

11. Figure 3b legend, it should be noted that maltose, glycerol, and NaCl are highlighted. It took me a moment to figure out what was going on here given the figure and the existing information in the figure legend.

We think this pertains to Figure 3C and if so, we have clarified this in the figure legend.

12. Figures 3d and 3e. I could not figure out what the dash/minus symbols meant. Please add to legend.

The dash represents a significant phenotype. We have added this to the figure legend.

13. P9, most would not refer to a GWAS study as a 'QTL analysis.' The latter typically refers to a mapping study focused on a known pedigree, such as a controlled cross.

We have changed "QTL analysis" to GWAS.

14. P9, it is not clear what the percentages refer to. I would have expected they refer to the proportion of the associations, but that cannot be the case.

The fraction is the fraction of all tested variants. We have clarified this in the manuscript.

15. The discussion is somewhat superficial, mainly just recapping the results of the paper. I wondered if the discussion couldn't be built out a bit, connecting the paper more to the broader

literature and problems. Also, the wording at the end of the discussion could be improved because it is wordy, but also somewhat vague. See: 'Despite an overall lack of enrichment, our results suggest that interpretation of the impact of genetic variants using the gene deletion information available for a single genetic background is unlikely to be comprehensive. In summary our results suggest that interpretation of the impact of genetic variants on the phenotypes of individuals would likely need detailed gene-phenotype information in more genetic backgrounds than that of a model individual.'

We have expanded and revised the discussion also taking into account the other reviewer's concerns.

16. Apostrophes used instead of commas in numbers equal to or greater than 1,000 in some places, such as first section of Methods.

This was corrected across the manuscript.

Reviewer #3:

The paper by Galardini and colleagues is a timely analysis bearing on important fundamental questions about how often and why a genetic variant causes a phenotype only some of the time (incomplete penetrance) or to varying quantitative extents (variable expressivity). Two important sources of incomplete penetrance (effects of genetic background and changes in environment) are explored. The papers use deletion collections in four different yeast strains to estimate how many knockout effects are strain-dependent (when considering growth in different environments). They find many strain-dependent knockout effects, especially those that are specific to S288C. This study is interesting and systematically extends earlier work exploring background-dependent knockout effects in yeast and other organisms. However, I do have some concerns about the analysis and presentation, and overall I think the manuscript can be suitable for publication if major revisions are made.

We thank the reviewer for the positive remarks.

Major:

-The conclusion that 16-42% of deletion phenotypes change between pairs of strains is interesting, but I am somewhat skeptical of the analysis:

-Regarding the null model: When trying to find significant differences between the growth of two knockouts, a null model should take into account the expected variability in both knockout measurements. However, it seems to me that their null model only considers variability in one knockout effect (N_{sub}), and not both. If I am mistaken, this should be more clearly explained. If this is correct, the null model should be changed to include the estimated variance of the effect in both strain (i.e. a pooled variance estimate), otherwise differences will be over-called.

In this manuscript, there are two key scores for which variance is taken into account - the measurement of gene-deletions in each strain named S-score and a score for the difference in gene-deletion scores between pairs of strains. We have performed 2 batches of experiments and each batch has 4 replicates for each condition. The 4 replicates are used to calculate the gene-deletion scores (S-score) in each condition as described in Collins al. Genome Biology

2006. We then calculate the difference in S-score across pairs of genetic backgrounds and for this we estimate a null model of the expected difference in gene-deletions. So we think this process does take into account the expected variance of the gene-deletion score in each strain as well as the expected variance of the difference in scores across strains. As we describe below, we have now determine error rates for determining a difference in gene-deletion scores.

-Related to the above point, how were the comparisons made when one or both of the conditions were not measured in two batches? The authors state that only 12 conditions were measured in two batches, but 38 were measured in total. It may be possible for the authors to have an (imperfect) variability estimate if something was measured only one batch (e.g. between their internal controls), but I do not see this described anywhere.

For the calculations of the S-score we made use of the 4 replicates to estimate variance of determining the colony size and calculate the S-scores as described Collins et al. 2006. The two batches were then used to estimate the variability in the S-scores in order to quantify a change in the S-score. As the reviewer has suggested, the variance model in determining a change in S-score is not specific to a given condition but instead estimates the variance in the difference in gene deletion S-scores across pairs of strains taking into account the magnitude of the S-scores. We have now made clear that the null model is not specific to a given condition.

-Even though the nominal FDR for differential growth analysis is low, the assumption of the 'null' FDR model is that the variability between two replicates is normally distributed. However, some data artifacts are evident which raise concern about false positives (which can inflate estimates of variable knockout phenotypes). For example, in both Figure 1C and 1D, there are some genes which seem to have a negative S score in Replicate 1/Gene copy 1 but not in the other replicate. As a sanity check, the authors should apply their differential fitness calling method between the scores obtained in Batch 1 and Batch 2, or have some other kind of 'empirical' control for how many false differential knockout effects they expect.

We agree with the reviewer on this important point that was also raised by the 1st reviewer. We have described in the response to reviewer 1 two attempts to calculate empirical false discovery rate for determining changes in gene deletion phenotypes. The first approach makes use of genes that are spotted more than once in each array and for the second approach we have generated again several knock-outs and performed a smaller scale repeat of the screens for some conditions to estimate the false discovery rate taken into account potential issues that may arise from the process of generating the mutants. In both of these estimates we measured empirical error rates below 5% with the exception of estimate of changes in gene-deletion scores for the S288C strain for newly generated mutants (17.6% errors). We think this is due to the fact that the S288C library has been generated a longer time away and likely contains more errors than the recently generated libraries. For this reason we suggest that the larger rate of changes in gene deletions scores estimated for S288C compared with the other strains is most likely due to this higher false positive rate due to accumulation of genetic variation.

- Re: "Even though a part of the observed changes might be false positives, we are confident that the homogeneity in experimental conditions as well as excluding uncertain cases from the analysis (Methods) helps reducing these cases to a small number." An unsupported statement

of the authors' confidence is simply not enough here. This issue needs quantitative evaluation. I am especially concerned about false positives because so many knockout effect comparisons are made (e.g. between strains, between environments) - so even if the false positive rate is low, many comparisons might lead to at least one significant difference just by chance. For example, the rate at which significantly different phenotypes are observed can be estimated for replicate batches. Given an estimate of reproducibility for positive and negative phenotype observations, what fraction of genes would show a phenotype in only 1 of 4, 2 of 4, 3 of 4 or 4 of 4 replicates, even when the ground truth remains the same?

We have removed the sentence. As we described above we have now derived empirical false positive rates that are below 5% with the exception of errors measured for newly generated gene deletion strains for S288c (17.6% false positive).

-Analysis of genetic and physical interaction degree relies on BioGrid, and therefore is unsystematic and subject to the well-known ascertainment biases of the literature. This analysis should be limited to systematic datasets within evenly tested search spaces. For example, Costanzo et al Science 2016 within the space of query x array genes used for mapping genetic interactions. Similarly, it could be Yu et al Science 2008 for direct protein interactions (within the space of protein pairs tested) or Gavin et al 2006 for co-complex associations (for all pairs involving a protein that was used as bait). It should be noted that correlations between genetic interaction degree and pleiotropy and between genetic interaction degree and single-mutant fitness were shown previously in Costanzo et al Science 2010. Correlation between direct protein interaction degree and pleiotropy was shown in Yu et al Science 2008.

We agree with the concern raised and have performed an analysis of physical and genetic interactions that were obtained only from systematic studies by excluding studies that reported less than 2000 interactions. The result did not change, with effects and p-values being similar to the ones observed without this filtering (p-value and effect sizes of $1.19E-18$, 0.55 and $1.4E-13$, 0.49 compared with the original $1.00E-9$, 0.53 and $5.18E-11$, 0.49). The resulting figure of this control analysis is shown here:

We mentioned this control in the corresponding results section.

We also agree with the reviewer that the number of physical and genetic interactions has been linked with the number of phenotypes. We are showing here how the number of physical/genetic interactions is associated with changes in gene-deletion phenotypes. We have expanded the discussion section to include this added context to these cited references.

-One potential source of variability is that the KO was properly made in some strains but not others. Quality control should be performed on a random sample of genes for which variation across strain backgrounds has been seen, in which there is a careful PCR-based analysis to make sure that: 1) the deletion cassette is present; 2) the junctions between flanking genomic region and deletion cassette are as expected; and 3) the target gene is actually absent. The latter is the most critical, as a common scenario when knocking out genes with fitness effects is that the only deletions that can be contained were in cells for which the target locus was duplicated, so that the deletion cassette replaces one copy of the target gene but leaves another copy of the target gene intact.

As described above in the response to reviewer 1's concerns, we have generated 16 gene deletions de novo for each genetic background (total 64 newly made KO strains). These strains were all PCR verified and screened again in 10 conditions. As we described the similarity of the growth measurements when comparing these new strains with the strains present in the libraries are high with the exception of some strains for the S288C strain. We think these new experiments address this concern directly.

Minor:

-There are some discrepancies between how the experiment is presented in the main text and how it is described in the methods. Specifically, the methods mention three batches (one of them measuring only S288C in some conditions), but two batches are mentioned elsewhere. Which is correct?

There were two batches that were carried out as fully independent experiments with 4 biological replicates each. We have revised the manuscript accordingly.

-It is shown that high-exclusiveness genes have a higher negative genetic interaction degree, and higher protein protein interaction degree. However:

-Figure 2E should contain a statistical test

We have added the statistic results to the corresponding text.

-It is known that genetic interaction degree for a gene correlates with its single-deletion effect (Costanzo et al 2010, 2018). Perhaps a similar correlation exists with PPI degree from Biogrid. Given results in Figure 3A, the background set should be chosen to have the same distribution of single-deletion effects (i.e. same mean and standard deviation) as the high-exclusiveness genes to correct for this.

To address this concern we selected genes that show a significant S-score in at least 5% of all strain/condition pairs tested. This results in both sets of genes - those with most changes and the rest - having a similar number and standard deviation of phenotypes (mean: 0.079 vs 0.074, std-dev: 0.047 vs 0.023). For these two sets of genes we still observe that the genes with a high number of changes in phenotypes have a higher number of protein and genetic interactions:

Differences are still significant and with similar effect size. We have added this result as an additional control.

-In Figure 2C, it is unclear why there are different estimates for number of genes shared amongst 4 strains - only 4 strains were tested, so this should just be a single set of genes (i.e. those which did not show a difference for any comparison). Why are there four bars, and why are the numbers different?

These bars represent the proportion of gene deletions from a given strain that are observed in all other 3 strains. The total number of phenotypes in each focal strain varies.

-When performing the Fisher's exact test to calculate significant overlap between associations and knockouts, were the genes that did not have usable variants excluded from the calculation? In general, the overlap may be significant if the authors don't consider genes where finding an association is very unlikely (e.g. not enough natural variability), and I think it is fair to do this filtering

We performed the suggested analysis by excluding genes that were not close to a variant SNP in at least 10 strains (3176 filtered genes studied). We found no difference in the enrichments.

-All association and enrichment analysis should give not only a measure of significance (p- or q-value) but also an estimate of effect size (e.g., enrichment factor). For negative results, giving the actual P-value is preferable to just saying $P > 0.05$

We have added a few of the effect size estimates where appropriate.

Issues with text:

-The authors should be consistent about whether they use 's-score' or 'S-score' (I think the latter is more correct)

-Text in Figure 1A is not legible at the size provided

-Colour legend in Figure 2B is not legible

-Gene names should be italicized

-Deletion notation is incorrect - e.g. *pdr5*Δ (italicized), not ΔPDR5

-Cycloheximide is spelled incorrectly, and should not be capitalized

-PDR5 deletion is generally leads to multidrug sensitivity, not resistance

-In general, paper could use more copyediting as there are several other grammar/spelling/formatting mistakes as well (e.g. "indicating the robustness of this trends", "duel stress conditions", "4'889 KOS")

-The word "numerosity" (the discrete property of being numerous) should be replaced with a continuous or cardinal quantifier, like "size" or "number".

We thank the reviewer for a careful review. We have attempted to correct these issues in the text and figures.

Thank you for sending us your revised manuscript. We have now heard back from the two reviewers who were asked to evaluate your study. As you will see below, the reviewers think that the study has improved as a result of the performed revisions. However, reviewer #3 still raises some remaining concerns, which we would ask you to address in a second round of revision.

REFEREE REPORTS

Reviewer #1:

The authors have addressed my concerns.

Reviewer #3:

The manuscript has been substantially improved, but some important issues remain unaddressed. We regret that two of the issues now listed could have been caught in the first submission but were missed until now..

Major:

-By using independently-generated strains, the authors have obtained an empirical estimate of the false discovery rate. However, their analysis points out that 18% of freshly made S288C deletions yield different phenotypes between fresh deletions and existing deletions. This is likely because individual deletion strains in the S288C background have experienced additional propagation and therefore had additional opportunity to adapt to the presence of deletions. It should be clarified whether the choice of target genes for the experiment where deletion strains were remade was random, or dependent in any way on the results of the first screen. If random, then this suggests that the abstract's claim of differences at rates up to 40% is an artifact, and the truth may well be substantially less than that.

-Related to the last point, the revision suggests that "This higher rate of error for S288C is likely due to accumulated secondary or compensatory mutations in the S288C KO library, which is consistent with previous reports (Teng et al , 2013) . It is likely that these genomic differences account, at least partially, for the higher degree of strain specific phenotypes observed for S288C (Figure 2C) . It is likely that these genomic differences account, at least partially, for the higher degree of strain specific phenotypes observed for S288C (Figure 2C) ."

The last sentence is a bit misleading. The term "strain specific phenotypes" is used here as a shorthand for "strain background-specific phenotypes". While specific adaptations in individual deletion strains can lead to "strain specific phenotypes" these would NOT be "strain background-specific phenotypes". So rather than saying that adaptation is an explanation for the higher degree of strain [background] specific phenotypes observed for S288C, it is evidence that many of the observed differences are NOT background-specific. The manuscript would be improved by using language that more clearly differentiates truly strain-specific from strain background-specific phenotypes.

-Related to the above point, statements like at a "q-value below 0.01 (per strain) we detected 2.6%, 3.5%, 4.5%, 0% false positives" are problematic: Because a q-value should represent a false discovery rate, differences between the nominal q-value and the empirical estimates of false discovery suggest that the procedure used to estimate q-values was inaccurate. Perhaps the internal replicate differences should be used to set q-value rather than the approach currently described.

-It is re-assuring that the correlations with BioGrid interactions hold after excluding smaller-scale studies. However, this restriction till does not properly protect against ascertainment bias (e.g. studies which more intensively studies selected baits and preys). Authors should evaluate studies that studied genes evenly within the space that was studied (e.g. Costanzo et al 2010/2016, Yu et al 2008). Moreover, these evaluations should be restricted to the set of gene pairs that was evenly studied within Costanzo et al 2010, etc.

-An issue that was regrettably missed in the first review was the explanation for the met5 phenotype reversal (resistance to amino acid starvation in S288C vs sensitivity in the other conditions). The stated explanation was a positive interaction between met5 and met17 but no citation is given and I could find no evidence of this in the literature. Moreover, it must be that methionine was

supplemented for any S288C deletion strain to have survived this condition, and this led me to wonder whether methionine (and other auxotrophy-marker-related amino acids) were supplemented for the S288C phenotyping but NOT for the other strains. This in turn raised another (major) issue... -Supplementary Table 1, which ostensibly describes the media used for phenotyping, does not begin to approach the level of detail that would allow the reader to understand, much less replicate, the reported results. For each of the drug conditions, the concentration of drug is described but really nothing else. What were the amino acid starvation conditions? If some amino acids were supplemented, which and at what concentration and was it the same for all strain backgrounds? If not, are "strain-background-specific differences" confounded by media differences?

Minor:

-Supplementary Figure 6 is missing

-Some previously-suggested copyediting still needs to be done - e.g. Cycloheximide vs cycloheximide. Others are also present - e.g. "Yest, deleting this gene"

We have addressed the issues brought up in this second round of revision and made minor changes to the manuscript text as described in the responses below. Our responses are in blue.

Referee comments:

The manuscript has been substantially improved, but some important issues remain unaddressed. We regret that two of the issues now listed could have been caught in the first submission but were missed until now..

Major:

-By using independently-generated strains, the authors have obtained an empirical estimate of the false discovery rate. However, their analysis points out that 18% of freshly made S288C deletions yield different phenotypes between fresh deletions and existing deletions. This is likely because individual deletion strains in the S288C background have experienced additional propagation and therefore had additional opportunity to adapt to the presence of deletions. It should be clarified whether the choice of target genes for the experiment where deletion strains were remade was random, or dependent in any way on the results of the first screen. If random, then this suggests that the abstract's claim of differences at rates up to 40% is an artifact, and the truth may well be substantially less than that.

Our analysis suggests that 18% of the gene-condition interaction changes in S288C may be due to genetic drift, not that 18% of the deletions give different phenotypes. If we were to remove 18% of the total number of changed interactions this would still result in 37.5% of changes observed between S288C and the other strains on average, compared to an average of 42% observed with the full dataset. Nevertheless, we agree with the reviewer that it is worth revising the abstract to account for the differences in S288C that are due to genetic changes in the S288C library. Instead of reporting the range of values we now report in the abstract the median value as the typical value. We added a sentence in discussion some with what we would expect could be the degree of changes for S288C if we were to account for the 18% error.

-Related to the last point, the revision suggests that "This higher rate of error for S288C is likely due to accumulated secondary or compensatory mutations in the S288C KO library, which is consistent with previous reports (Teng et al , 2013) . It is likely that these genomic differences account, at least partially, for the higher degree of strain specific phenotypes observed for S288C (Figure 2C) . It is likely that these genomic differences account, at least partially, for the higher degree of strain specific phenotypes observed for S288C (Figure 2C) ."

The last sentence is a bit misleading. The term "strain specific phenotypes" is used here as a shorthand for "strain background-specific phenotypes". While specific adaptations in individual deletion strains can lead to "strain specific phenotypes" these would NOT be "strain background-specific phenotypes". So rather than saying that adaptation is an explanation for the higher degree of strain [background] specific phenotypes observed for S288C, it is evidence that many of the observed differences are NOT background-specific.

The manuscript would be improved by using language that more clearly differentiates truly strain-specific from strain background-specific phenotypes.

We have revised the sentence accordingly. We now state that this result suggests that on the order of 18% of gene-condition changes estimated for S288C could be errors. As discussed below, even an 18% error rate on calling a change in gene-deletion phenotype would not strongly impact on the overall rate of change.

-Related to the above point, statements like at a "q-value below 0.01 (per strain) we detected 2.6%, 3.5%, 4.5%, 0% false positives" are problematic: Because a q-value should represent a false discovery rate, differences between the nominal q-value and the empirical estimates of false discovery suggest that the procedure used to estimate q-values was inaccurate. Perhaps the internal replicate differences should be used to set q-value rather than the approach currently described.

It is an unrealistic expectation that an estimate of false discovery based on multiple testing theory and an empirical measurements of false discovery will exactly match. The fact that the theoretical value (0.01) approximates the observed values (0.026, 0.035, 0.045 and 0) is certainly within the expectation here.

-It is re-assuring that the correlations with BioGrid interactions hold after excluding smaller-scale studies. However, this restriction still does not properly protect against ascertainment bias (e.g. studies which more intensively studied selected baits and preys). Authors should evaluate studies that studied genes evenly within the space that was studied (e.g. Costanzo et al 2010/2016, Yu et al 2008). Moreover, these evaluations should be restricted to the set of gene pairs that was evenly studied within Costanzo et al 2010, etc.

There is no expectation as to why the large scale studies that we have selected would be biased towards "more intense study" of genes that were found to have the largest number of changes in gene deletion phenotypes. In fact, no obvious reason as to why these systematic studies will have a strong bias in the degree of analysis of different genes either. Still, we have performed the analysis suggested by the reviewer. We have restricted the BioGrid database to the following Pubmed IDs: 27708008 and 20093466 (for genetic interactions) and 18719252 (for physical interactions). For each respective analysis we considered only genes that had at least 1 interaction in the genetic or physical interaction data to guarantee that the genes were sampled in the interaction studies. We then repeated the analysis presented in Figure 2E:

Both comparisons are significant (KS test p-value 1.6E-9 and 3.9E-8 and Cohen's d 0.55 and 0.50, respectively), which again is very similar to the two previous analyses.

-An issue that was regrettably missed in the first review was the explanation for the met5 phenotype reversal (resistance to amino acid starvation in S288C vs sensitivity in the other

conditions). The stated explanation was a positive interaction between met5 and met17 but no citation is given and I could find no evidence of this in the literature. Moreover, it must be that methionine was supplemented for any S288C deletion strain to have survived this condition, and this led me to wonder whether methionine (and other auxotrophy-marker-related amino acids) were supplemented for the S288C phenotyping but NOT for the other strains. This in turn raised another (major) issue...

The explanation that we provided in the text was a hypothesis, a speculation. As we wrote “a potential explanation for this phenotype reversal could be a positive genetic interaction between MET5 and MET17”. We have revised this sentence to make it hopefully clearer that this is an unproven hypothesis. As per the methods description, there were no differences in the media used between the strains.

-Supplementary Table 1, which ostensibly describes the media used for phenotyping, does not begin to approach the level of detail that would allow the reader to understand, much less replicate, the reported results. For each of the drug conditions, the concentration of drug is described but really nothing else. What were the amino acid starvation conditions? If some amino acids were supplemented, which and at what concentration and was it the same for all strain backgrounds? If not, are "strain-background-specific differences" confounded by media differences?

The media that used was the same for all strains as described in the methods section: "Synthetic complete (Kaiser et al, 1994) media was used with or without the stress condition". We have added in supplementary table 1 the amino-acids and concentrations used for the amino acid starvation condition

Minor:

-Supplementary Figure 6 is missing

We have added the missing figure.

-Some previously-suggested copyediting still needs to be done - e.g. Cycloheximide vs cycloheximide. Others are also present - e.g. "Yest, deleting this gene"

We have made these suggested corrections.

Thank you again for sending us your revised manuscript. We are now satisfied with the modifications made and I am pleased to inform you that your paper has been accepted for publication.

Corresponding Author Name: Pedro Beltrao

Journal Submitted to: MSB

Manuscript Number: MSB-19-8831R